# Mitigating Unintended Memorization with LoRA in Federated Learning for LLMs

**Thierry Bossy**[*]
*Tune Insight SA, Switzerland*

*thierry@tuneinsight.com*

**Julien Tuấn Tú Vignoud**[*]
*EPFL, Switzerland*

*julien.vignoud@epfl.ch*

**Tahseen Rabbani**
*Yale University, USA*

**Juan R. Troncoso**
*Tune Insight SA, Switzerland*

**Martin Jaggi**
*EPFL, Switzerland*

**Reviewed on OpenReview:** *https://openreview.net/forum?id=WKPyZnLIW4*

## Abstract

Federated learning (FL) is a popular paradigm for collaborative training which avoids direct data exposure between clients. However, data privacy issues still remain: FL-trained large language models are capable of memorizing and completing phrases and sentences contained in training data when given their prefixes. Thus, it is possible for adversarial and honest-but-curious clients to recover training data of other participants simply through targeted prompting. In this work, we demonstrate that a popular and simple fine-tuning strategy, low-rank adaptation (LoRA), reduces memorization during FL by a factor of up to 10 without significant performance cost. We study this effect by performing fine-tuning tasks in high-risk domains such as medicine, law, and finance. We observe a reduction in memorization for a wide variety of model families, from 1B to 70B parameters. We find that LoRA can reduce memorization in centralized learning as well, and we compare how the memorization patterns differ. Furthermore, we study the effect of hyperparameters and show that LoRA can be combined with other privacy-preserving techniques such as gradient clipping and Gaussian noise, secure aggregation, and Goldfish loss to further improve record-level privacy while maintaining performance.

## 1 Introduction

Large language models (LLMs) have been shown to achieve state-of-the-art performance over most relevant natural language processing (NLP) tasks (Zhao et al., 2023). Following these advances, there is significant interest in fine-tuning LLMs for downstream tasks over specialized domains such as medicine (Singhal et al., 2023a; Thirunavukarasu et al., 2023; Yang et al., 2022), law (Cui et al., 2024a) or finance (Wu et al., 2023b; Li et al., 2023b). Given the inherent confidentiality of user data involved in these fields, as well as the increasing number of studies showing LLMs' propensity to expose training data (Nasr et al., 2025; Carlini et al., 2023a; Hou et al., 2025; Hayes et al., 2025; Leybzon & Kervadec, 2024; Zeng et al., 2024; Biderman et al., 2023), there is a crucial need for additional privacy mechanisms. One such mechanism for data-constrained parties is federated learning (FL), a widely-studied paradigm enabling multiple data parties, called *clients*, to

---

[*]Equal contribution

collaboratively train a machine learning model without sharing local data (McMahan et al., 2016; Kairouz et al., 2021).

An early study by Thakkar et al. (2021) of a 1.3M parameter LSTM next-word predictor (Hard et al., 2019) showed that FL significantly reduces unintended memorization compared to centralized learning (CL). Yet, it is unclear whether FL remains effective at preventing recent multi-billion parameter models with Transformers architectures (Vaswani et al., 2023) from memorizing training data and exposing sensitive information at inference time.

Moreover, there is a significant interest in the FL community in leveraging parameter-efficient fine-tuning (PEFT) techniques (Kuang et al., 2023; QI et al., 2024; Wu et al., 2024; Yi et al., 2024; Sun et al., 2024; Liu et al., 2024), and in particular Low-Rank Adaptation (LoRA) (Hu et al., 2021). Indeed, LoRA and other PEFT techniques offer multiple desirable properties. By only updating a small set of parameters while keeping the pre-trained weights frozen, PEFT methods alleviate the prohibitive computational costs of training models comprised of multiple billion parameters and drastically reduce the size of the updates exchanged during the FL training. While the impact of LoRA on unintended memorization is gaining interest in centralized learning (Hou et al., 2025; Wang & Li, 2025), its specific effects in FL are not well-understood yet and mainly studied in combination with differential privacy (DP) (Dwork et al., 2006) by Sun et al. (2024) and Liu et al. (2024). Both works evaluated the performance of LoRA in FL combined with DP and found non-trivial performance losses even for high privacy budgets.

In this paper, we extensively evaluate the resulting unintended memorization of FL-trained LLMs in a data-heterogeneous 3-client setup and show that LoRA enables significant memorization reduction for little to no performance cost compared to full fine-tuning. We fine-tuned models over realistic sensitive information and measure the rate of unintended memorization in different domains such as medicine, law and finance. In contrast to Thakkar et al. (2021), we extend our analysis to LoRA fine-tuning, and study memorization across a broader range of settings, model families and model sizes up to 70B parameters. Our work is complementary to Sun et al. (2024) and Liu et al. (2024): we empirically measured memorization rather than under DP theoretical bounds, via several memorization metrics such as exact token matching, approximate reproduction Ippolito et al. (2023) and BERTScore (Zhang et al., 2020). While DP provides theoretical bounds, we argue that a complementary empirical evaluation is essential. Indeed, DP's applicability to LLM training on natural language data and its subsequent formal guarantees have been questioned by Brown et al. (2022) and Tramèr et al. (2024), as it remains unclear how DP's *secret boundaries* should be defined given the contextual nature of language and in the light of successful personally identifiable information (PII) extraction of DP-trained LLMs (Lukas et al., 2023).

Our contributions are as follows:

- We empirically demonstrate that LoRA mitigates memorization in federated learning for little to no performance cost compared to full fine-tuning. This effect generalizes to datasets drawn from several sensitive data domains such as medicine, law, and finance.

- We comprehensively test model sizes from 1B to 70B parameters from the Llama-2 family, Llama-3 family, and Mistral-v0.3. LoRA effectively reduces memorization across models.

- We investigate the impact of the LoRA rank on memorization and compare how sensitive data memorization in federated learning differs from centralized learning.

- We experimentally explore how LoRA interacts with other privacy strategies. This includes differential privacy mechanisms such as gradient noising and clipping, Goldfish loss (Hans et al., 2024), post-training noise injection and secure aggregation. We demonstrate how LoRA can work synergistically with these other approaches.

- We release a repository with code and instructions to reproduce our results: `https://github.com/tuneinsight-collab/FederatedLLMs`.

# 2 Related Work and Preliminaries

This section reviews related work and foundational concepts in LoRA, federated learning, and memorization in large language models. Additional discussions on differential privacy, membership inference attacks, secure aggregation, and medical applications are provided in Appendix A.

## 2.1 LoRA

To reduce computational and memory requirements when fine-tuning LLMs, Low-Rank Adaptation (LoRA) (Hu et al., 2021) was introduced to drastically reduce the number of trainable parameters during fine-tuning. This is achieved by representing the weight updates $\Delta W$ as the product $\Delta W = BA$ of two low-rank matrices $A$ and $B$. LoRA enables efficient adaptation of LLMs to specific tasks while preserving the generalization capabilities of the underlying model, as gradients often exhibit low intrinsic dimensionality (Li et al., 2018; Aghajanyan et al., 2020). Additionally, LoRA offers a notable advantage in an FL scenario by drastically reducing the amount of data exchanged between participants during each round. In our experiments, we achieved a 130-fold reduction. Understanding LoRA's resultant performance compared with full fine-tuning is still an active area of research, with preliminary results showing LoRA can match full model training on small to medium-sized datasets (Schulman & Lab, 2025; Biderman et al., 2024).

The study of LoRA and memorization has been limited to centralized learning, mainly by Hou et al. (2025) and Wang & Li (2025), and indirectly by Biderman et al. (2024). Hou et al. (2025) compares memorization between LoRA and prompt-based fine-tuning, which consists in keeping the pre-trained model weights frozen and learning task-specific prompts. However, their study does not include a comparison with full parameter fine-tuning, which we argue is most widely used in practice along with LoRA. Recent work by Wang & Li (2025) finds that fine-tuning GPT-2 with LoRA results in lower memorization than full fine-tuning, suggesting its potential for federated learning. Biderman et al. (2024) analyzes the trade-off between factual knowledge learning and forgetting. While they do not directly analyze training data memorization, they show that LoRA presents a better factual knowledge "learning-forgetting" tradeoff than full fine-tuning, learning less on math and code datasets but forgetting less of its pre-training knowledge.

## 2.2 Federated Learning

Federated learning (FL) has been widely-studied for deep learning models in cross-silo settings Huang et al. (2022), in which a limited number of resource-rich clients, such as organizations or institutions, collaboratively train ML models without sharing their data. In conventional FL, the global objective function of $N$ clients is defined as

$$\min_{W} F(W) = \sum_{k=1}^{N} p_k f_k(W), \tag{1}$$

where $W$ represents the parameters of a model, $\sum_{k=1}^{N} p_k = 1$ and $f_k(W)$ is the local objective function of client $k$. Local training data $\mathcal{D}_k$ between clients is often heterogeneous. A common strategy for solving Equation 1 is Federated Averaging (FedAvg) (McMahan et al., 2016). FL has recently been applied to LLMs Ye et al. (2024); Thakkar et al. (2021); Liu et al. (2024); Ramaswamy et al. (2020) leveraging FedAvg to aggregate locally-trained model updates. Work by Thakkar et al. (2021) informed our federated training strategy by demonstrating on a small scale that federated averaging is especially effective at reducing memorization on non-independent and identically distributed (non-IID) data. Recent work by Google and others has explored the use of FL for large-scale language model training in production environments, placing strong emphasis on privacy protection (Hard et al., 2019; Ramaswamy et al., 2020; Thakkar et al., 2021; Xu et al., 2023), showing growing interest in privacy-preserving training methods. While Liu et al. (2024) and Sun et al. (2024) studied LoRA with DP-SGD, this work is to the best of our knowledge the first to empirically measure the impact of LoRA on memorization in federated learning.

### 2.3 Memorization

How to quantify the memorization capacity of an LLM is an active area of research. A seminal work by Carlini et al. introduced "canaries", which are synthetic, out-of-distribution pieces of text injected into training data (such as `"My SSN is XXX-XX-XXXX"`) (Carlini et al., 2019). It has found use in production-level studies (Ramaswamy et al., 2020) and adjacent fields such as machine unlearning (Jagielski et al., 2022). An alternative definition of memorization (Carlini et al., 2023a), the completion metric, measures how often an LLM completes a piece of text taken from the training data when prompted on an initial portion (prefix) of it.

**Memorization definition.** We use the "extractable memorization" definition of Carlini et al. (2023b) following its wide adoption (Ippolito et al., 2023; Huang et al., 2024; Hans et al., 2024; Nasr et al., 2023; 2025). Consider a string representable as the concatenation $[p||s]$ where $p$ is a prefix of length $k$ and $s$ is the remainder of the string. We define the string $s$ to be *memorized with k tokens of context* by a language model $f$ if $[p||s]$ is contained in the training data of $f$, and $f$ produces $s$ when prompted with $p$ using greedy decoding[1]. In other words, we consider a string from training data memorized if an LLM can generate it when prompted by a prefix. We set the length of the generated suffix $s$ to 50 tokens, in line with previous work.

## 3 Methodology

In a federated learning setting, training data is split among several clients. Our experiments are designed to mimic a medical scenario in which each client holds potentially sensitive data, reflecting the reality that few, if any, anonymization tools can guarantee the complete removal of sensitive information (Langarizadeh et al., 2018; Brown et al., 2022). In fact, Heider et al. (2020) evaluated three off-the-shelf de-identification tools on the i2b2 medical record dataset (Stubbs & Özlem Uzuner, 2015)—which we use in our study—and found that none could achieve complete removal. We adopt a so-called *cross-silo* setting, where there are only a handful of participants, 3 in our case, each with a relatively large amount of data, in contrast to *cross-device* FL in which data is divided among up to millions of clients. Cross-silo FL is a realistic setting for a medical scenario. For example, a few hospitals with confidentiality requirements may want to collaborate and train a more generalizable model. Moreover, a large number of clients is computationally prohibitive for the model scale we are studying in this work.

### 3.1 Quantifying memorization

Our measurement methodology is largely inspired by Carlini et al. (2023b). In short, we inject sensitive sequences, so-called "canaries" (Carlini et al., 2019; Jagielski et al., 2023; Thakkar et al., 2021), into fine-tuning data and then measure the model's ability to regurgitate this information when prompted with the beginning of these sequences.

**Canaries.** Unlike prior work that evaluates the memorization of all training data (Carlini et al., 2023b; Ippolito et al., 2023; Hans et al., 2024), we are interested in measuring how much *sensitive* information is memorized. Similar to Lehman et al. (2021) and Mireshghallah et al. (2022), we inject medical records into our training set originating from the 2014 i2b2/UTHealth corpus (Stubbs & Özlem Uzuner, 2015). The i2b2 dataset contains 1,304 longitudinal medical records that describe 296 patients. This approach allows us to measure memorization on sensitive data that we explicitly want to protect, while most previous studies measure memorization on randomly-generated sequences (Carlini et al., 2019; Thakkar et al., 2021) or indiscriminately on the whole dataset (Wang & Li, 2025; Biderman et al., 2024; Hans et al., 2024; Nasr et al., 2025) possibly confounding the memorization necessary for downstream performance Feldman (2020).

Because data duplication has been shown to greatly influence memorization (Carlini et al., 2023b; Lee et al., 2022; Kandpal et al., 2022), we randomly select 30% of the medical records and duplicate them tenfold within our fine-tuning data in order to study the effect of data duplication in our experiments. We also experiment with a lower duplication rate of 3 in Appendix C and found consistent results. Following Carlini et al. (2023b),

---

[1]Carlini et al. (2023a) found that beam search yields slightly higher memorization values. We use greedy decoding for the sake of simplicity.

we measure the effect of the context size by prompting the model on each test sequence with prompts of lengths in $\{10, 50, 100, 200, 500\}$. The different prompts for a given test sequence are constructed such that the suffix $s$ is kept identical while varying the prompt length. This ensures a fair comparison between prompt lengths, since different suffixes may be more or less prone to regurgitation.

**Memorization scores.** To compare generated text with the ground truth, we rely on two metrics: (1) the **exact token match rate** and (2) the **BLEU score**, to measure approximate reproduction, as prior work suggests that the exact match rate does not capture subtler forms of memorization (Ippolito et al., 2023). In line with this work, we consider a sequence memorized if the generated suffix and the ground truth yield a BLEU score $> 0.75$. For both metrics, lower is better, and a score of 1 denotes complete memorization of all test sequences. We additionally report **BERTScore** (Zhang et al., 2020) in Appendix C to measure semantic similarity between model outputs and reference sequences and we find trends consistent with the aforementioned metrics.

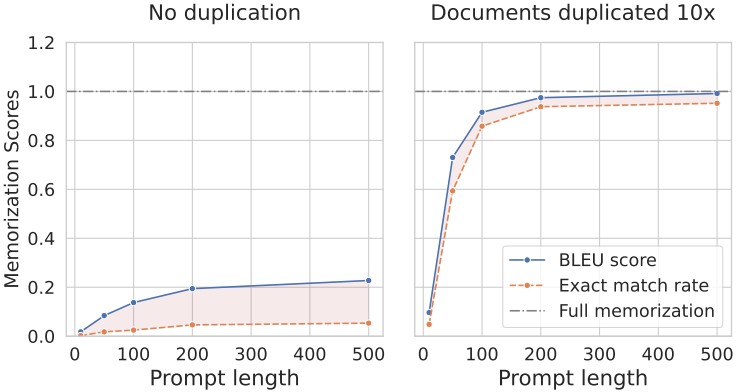

Figure 1: **Sensitive medical information memorization of the full fine-tuning of Llama 2 7B in centralized learning.** We report the exact match rate and BLEU score with respect to the prompt length, with and without duplication. We also show the memorization upper bound (labeled "Full memorization") that is reached when every test sequence has been memorized. Most settings show signs of regurgitation yet duplicated documents present an alarming rate of memorization.

To illustrate our method, Figure 1 shows the memorization of the Llama 2 7B full fine-tuning. Multiple trends are consistent with previous work: (1) there is significantly more memorization when the medical records occur multiple times in the fine-tuning data (Lee et al., 2022; Kandpal et al., 2022; Carlini et al., 2023b); (2) longer prompts show higher memorization (the so-called "discoverability phenomenon") (Carlini et al., 2023b) and (3) there is significantly more memorization with approximate generation (BLEU score) (Ippolito et al., 2023).

**Accuracy.** We report the downstream accuracy in Appendix B to ensure a fair comparison between fine-tuning methods and ensure that potential privacy gains do not come at the expense of decreased performance. Figure 6 shows that all fine-tuned models yield relatively similar accuracy values when comparing LoRA to full fine-tuning. This result suggests that in our setting, LoRA is a competitive technique and may substitute full fine-tuning at relatively little cost.

## 3.2 Experimental Setup

**Datasets and models.** We fine-tune LLMs on three medical QA datasets (MedMCQA, PubMedQA, and Medical Meadow Flashcards) augmented with sensitive sequences from the i2b2 clinical notes (Stubbs & Özlem Uzuner, 2015). Evaluation is performed on a suite of medical benchmarks including MedQA, PubMedQA, MedMCQA, and MMLU-Medical (Pal et al., 2022; Jin et al., 2019; Han et al., 2023). Full descriptions of datasets, pre-trained models, and licensing terms are provided in Appendix E. In Appendix C, we confirm that our findings generalize to other high-risk domains such as law (Multi-LexSum) and finance (ConvFinQA) (Shen et al., 2022; Cheng et al., 2024) and to a larger model scale of 70B.

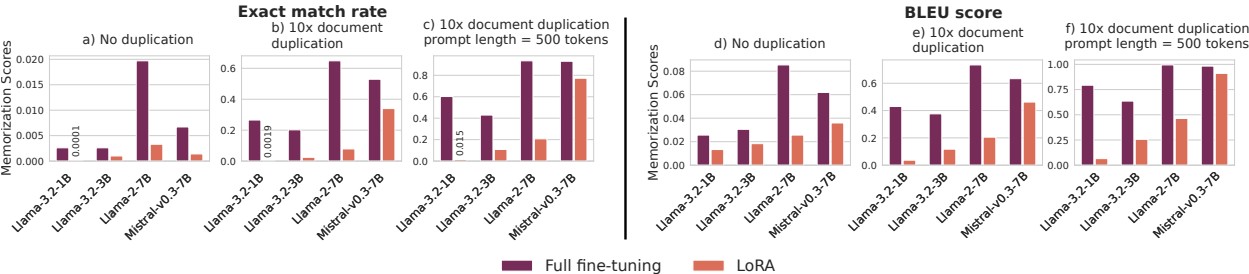

Figure 2: **LoRA vs. full fine-tuning in *federated* learning.** LoRA results in lower unintended memorization than full fine-tuning in FL across all settings. (a)–(c): Exact match rate under increasing duplication and prompt length. (d)–(f): BLEU score under the same settings. The memorization scores are averaged across the 5 different prompt lengths (from 10 tokens to 500 tokens). (c) and (d) show our worst case setting by using the highest prompt length (500 tokens) and a duplication rate of 10. As control, we evaluated the memorization scores before fine-tuning and found values lower than 0.0006 for the exact match rate (i.e.,less 0.06% of the tested sequences resulted in a verbatim regurgitation) and BLEU scores lower than 0.003.

**Cross-silo FL.** We fine-tuned models with three participants, where each participant trained locally on one of the three datasets MedMCQA, PubMedQA, and Medical Meadow Flashcards in a heterogeneous data distribution. Previous work showed the effectiveness of non-IID data at mitigating unintended memorization (Thakkar et al., 2021), therefore adopting a heterogeneous setting is a memorization-wise best-case scenario compared to an IID setting. We split and injected i2b2 medical records into each dataset proportionally to their size. Participants fine-tuned over their local dataset for one epoch between each global weight update, for a total of five rounds. For every model, we tuned the learning rate separately on each local dataset. To better understand the privacy impact of FL itself, we compared the same experiments in conventional centralized learning in Section 4.1, where all training samples were processed by a single participant.

All experiments were performed on a single NVIDIA A100 80GB GPU within an HPC cluster, except for the 70B-parameter model, which was fine-tuned using eight H100 GPUs. We leveraged Hugging Face's Transformers library (Wolf et al., 2020) to access and fine-tune pre-trained models. Further training details are included in Appendix D.

## 4 Results

Figure 2 compares the impact on memorization of replacing full fine-tuning by LoRA in FL. **Fine-tuning federated LLMs with LoRA results in lower unintended memorization than full fine-tuning across all metrics and models**. In our experiments, LoRA fine-tuning reduced memorization up to $10\times$ for a negligible accuracy loss, as shown in Figure 6.

Notably, data duplication and longer prompt lengths greatly increases unintended memorization, extending these trends from CL (Lee et al., 2022; Carlini et al., 2023a;b) to FL. Also in line with previous work, smaller models exhibit less memorization than larger ones, though we found that Llama 2 7B and Mistral v0.3 7B showed different memorization dynamics despite having the same size. For example, fine-tuning Llama 2 7B with LoRA show a drastic memorization improvement over full fine-tuning, whereas LoRA has a lower impact with Mistral v0.3 7B. Conversely, Llama 2 7B displays a higher memorization rate than Mistral v0.3 7B in full fine-tuning. We discuss this result further in the Section 4.1 by comparing FL and CL results.

Additionally, we found that full fine-tuning in FL still results in alarmingly high rates of memorization despite the privacy-enhancing properties of FL observed by Thakkar et al. (2021). This result emphasizes the limits of FL as a privacy-preserving method. Furthermore, even LoRA exhibits some levels of memorization, despite being lower, thus showing the need for additional privacy-preserving techniques.

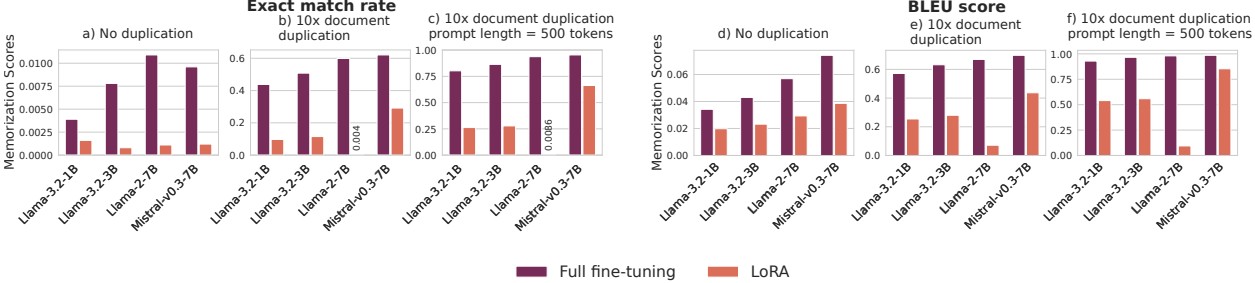

Figure 3: **LoRA vs. full fine-tuning in *centralized* learning.** LoRA consistently reduces unintended memorization compared to full fine-tuning. (a)–(c): Exact match rate under increasing duplication and prompt length. (d)–(f): BLEU score under the same settings. We obtained pre-fine-tuning controls scores an order of magnitude lower than any fine-tuned model score, which additionally confirms that none of the models had already been trained on the i2b2 dataset. While some scores appear low at first glance, the lowest memorization depicted in this figure remains >10 times higher than the control.

To provide fair comparisons between multiple federated learning fine-tuning, Figure 2 reports metrics for the last federated round. This ensures that each model has been fine-tuned on the medical records the same number of times. We discuss this decision further in Appendix B.

## 4.1 LoRA in FL vs CL

To differentiate the effects of LoRA and FL on memorization, we carried out the same experiments in centralized learning. In the CL setting, we merged PubMedQA, MedMCQA and Medical Meadow Flashcards into one fine-tuning dataset, in which we injected the i2b2 medical records to benchmark memorization after fine-tuning. We used a validation split of 10% and for each model we searched for the learning rate yielding the lowest validation loss. More details on hyperparameters can be found in Appendix D.

Figure 3 shows that **models fine-tuned in CL with LoRA consistently exhibits lower memorization scores than full fine-tuning**, suggesting the adequacy of using LoRA as a memorization-mitigating technique in both FL and CL. Across all model sizes, data duplication and longer prompt lengths greatly increase memorization. The figure also illustrates that larger models memorize more (Carlini et al., 2023b; Tirumala et al., 2022). Contrary to our results, Wang & Li (2025) found that larger models and data duplication in CL did not affect memorization LoRA fine-tuning, with memorization scores remaining near zero for all settings while full fine-tuning memorization was increasing. We suggest that near-zero memorization can result from the small scale of GPT-2 models, and that memorization may appear only above a certain model size. In fact, the two trends do appear when they relax their definition of memorization, resulting in Llama 3 1B and 8B showing non-trivial memorization with LoRA.

Comparing memorization scores between Figure 2 and 3, we found that FL itself enhances privacy by reducing memorization compared to CL for a given fine-tuning method. This is consistent with previous work by Thakkar et al. (2021) who suggested that FedAvg and a non-IID data distribution contribute to reducing unintended memorization. Furthermore, we found that not all trends observed in FL hold in CL. On the one hand, data duplication, longer context and considering paraphrasing all yield higher memorization scores for FL as for CL, on the other hand, Figure 2 shows that larger models do not necessarily result in more memorization with full fine-tuning in FL, as Llama 3.2 1B reached higher memorization scores than Llama 3.2 3B. This difference may stem from FL using separate learning rates for each local model, and thus each data source, while in centralized learning a single learning rate is used (since all datasets are merged into one). This finer-grained adaptation in FL, particularly in the non-IID setting where clients train on domain-specific data, can also explain the slightly higher memorization observed when training Llama 2 7B with FL compared to CL.

Interestingly, as in FL, LoRA is less effective at reducing memorization for Mistral v0.3 7B than for Llama 2 7B in CL as well. We hypothesize that architectural features may cause different memorization dynamics.

Notably, Mistral v0.3 leverages a sliding window attention (Beltagy et al., 2020) with Grouped-Query Attention (Ainslie et al., 2023) while Llama 2 7B uses a regular multi-head attention. We leave further analysis of the impact of architectural components on unintended memorization for future work.

To better understand the memorization dynamics in FL, we measured memorization with LoRA and full fine-tuning with respect to federated rounds, as shown in Figure 4. As expected, we found that memorization in FL increases monotonically with the number of rounds (which corresponds to the number of times medical records are seen). Throughout the rounds, we see that LoRA memorizes less than full fine-tuning, with some exceptions where LoRA starts with slightly higher memorization in the first or second round. The choice of the number of rounds is thus critical, and we show in Table 3 that full fine-tuning reached its best downstream accuracy in the last two rounds, where the memorization gap between LoRA and full fine-tuning is the greatest. In other words, LoRA is most effective at reducing memorization where the models are most performant.

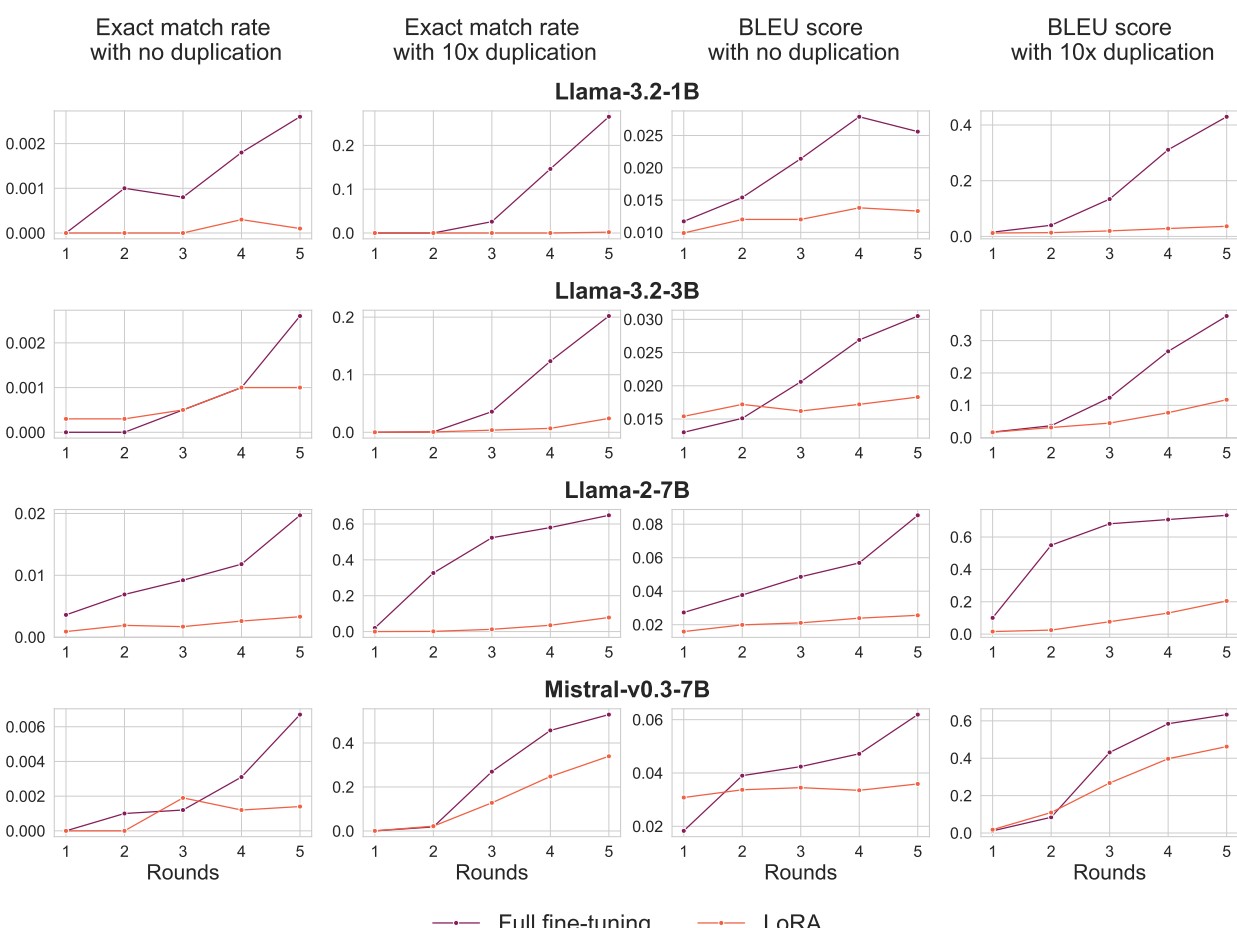

Figure 4: **Memorization evolution throughout the federated learning rounds.** The FL memorization values found in Figure 2 are the values at the last round in this Figure.

To summarize our results, we found that **combining LoRA with FL synergistically mitigates unintended memorization across varying model sizes and duplication rates**. Furthermore, we show in Appendix C that these results generalize to other domains such as law and finance, to lower duplication rates, as well as to larger models such as Llama 3.1 70B in CL. Future work is necessary to understand differing memorization dynamics between similarly-sized models, for example by exploring the effects of differing attention mechanisms.

## 4.2 The privacy-utility tradeoff

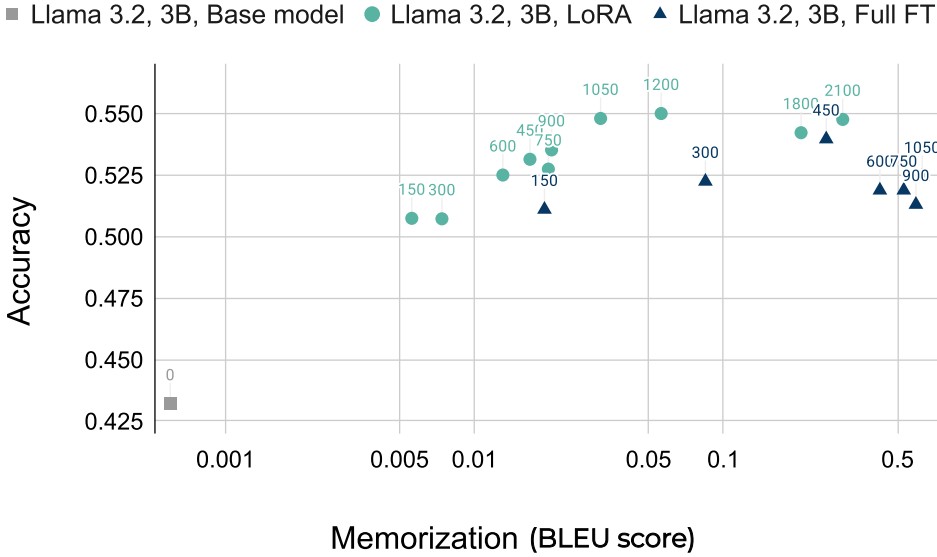

Figure 5: **Accuracy vs. privacy across fine-tuning steps.** We tracked the accuracy and memorization (BLEU) during the Llama 3.2 3B fine-tuning (10× duplication) using full fine-tuning (Full FT) and LoRA, compared to the base model. Numbers above points indicate completed fine-tuning steps.

To assess whether the privacy gains resulting from replacing full fine-tuning with LoRA is due to overfitting and is preventable by early stopping, we analyzed the utility-privacy tradeoff throughout the CL fine-tuning steps. Comparing utility and privacy with respect to training steps further allows us to assess whether the privacy gains observed when training models with LoRA and CL come at the cost of utility. Figure 5 illustrates the evolution of the privacy and utility for Llama 3.2 3B for LoRA and full fine-tuning. The figure shows that **LoRA consistently follows a more privacy-preserving trend than full fine-tuning throughout the training steps**, with lower memorization scores at similar utility levels. Furthermore, after a certain number of fine-tuning steps, the model's tendency to memorize data increases without significant improvements in utility, due to overfitting. This highlights that early stopping during LLM training not only improves efficiency, but also reduces the risk of memorization.

Importantly, LoRA even achieves slightly better peak accuracy than full fine-tuning. This can stem from LoRA's inherent resistance to overfitting, as restricting updates to a low-rank subspace acts as a form of regularization (Biderman et al., 2024). In contrast, full fine-tuning begins to overfit after several hundred steps, which limits its attainable accuracy. Additional regularization could likely improve full fine-tuning.

## 4.3 The LoRA rank and memorization

We further investigated how LoRA's configuration influences unintended memorization. Specifically, we varied the LoRA rank to measure its influence on memorization, studying values $r \in \{4, 16, 64, 128, 256, 1024\}$. The scaling factor $\alpha$ is set to twice the rank, as recommended by Biderman et al. (2024), and the learning rate is decreased exponentially as the rank increases. We applied LoRA adapters to all layers, following Biderman et al. (2023) and Schulman & Lab (2025).

As shown in Table 1, increasing the rank—i.e. increasing the number of weights updated during fine-tuning results in more memorization, ranging from virtually no memorization with a rank of 4 to almost 50% of the medical records being memorized for rank 1024 when considering duplicated medical records. These results are consistent with Wang & Li (2025) and extend their conclusions to a larger model than GPT-2. Note that in our case, the highest rank did not yield the best accuracy. We further note that for most ranks,

Table 1: **Impact of the LoRA rank on memorization.** We fine-tuned Llama 3.2 3B with LoRA in centralized learning on increasing LoRA ranks. Higher ranks lead to more memorization but not necessarily to better accuracy. The lower memorization scores and the highest accuracy are emphasized in bold.

| LoRA rank | Exact match rate | | BLEU score | | Accuracy |
|---|---|---|---|---|---|
| | No duplication | 10x duplication | No duplication | 10x duplication | |
| 4 | 0.0003 | 0 | 0.0133 | 0.0198 | 0.509 |
| 16 | 0.0005 | 0.0031 | 0.0167 | 0.0623 | 0.512 |
| 64 | 0.0031 | 0.2105 | 0.0258 | 0.379 | 0.511 |
| 128 | 0.0042 | 0.3735 | 0.0305 | 0.5111 | 0.510 |
| 256 | 0.0057 | 0.4895 | 0.0352 | 0.5809 | **0.542** |
| 1024 | **0.0063** | **0.4981** | **0.0409** | **0.6228** | 0.530 |

suboptimal learning rates can lead to relatively significant memorization with LoRA, though still lower than for full fine-tuning, highlighting the need for careful hyperparameter tuning.

### 4.4 Combining LoRA with other privacy-enhancing methods

Although LoRA mitigates unintended memorization on its own, we investigated whether it can be combined with other privacy-preserving techniques without compromising performance or increasing memorization. In this section, we outline the different techniques we evaluated, while detailed descriptions of each are provided in Appendices F, G, H, and J.

**Goldfish loss.** If users are focused on reducing extractable memorization in pretraining, then they may be interested in Goldfish loss, while LoRA is used for fine-tuning. However, we investigated its potential for fine-tuning in combination with LoRA in Appendix F and show that the combination of LoRA with Goldfish loss synergistically achieves lower memorization beyond what either strategy achieves alone.

**NEFTune.** We examine NEFTune in Appendix G, a noise-enhanced fine-tuning approach designed to improve robustness and reduce overfitting. While not a privacy-preserving method per se, we hypothesized that noise addition could induce some memorization reduction. In practice, we found that combining NEFTune with LoRA fine-tuning does not further decrease the memorization and in fact does not improve the performance either, the highest accuracy being reached without any noise. We hypothesize that NEFTune's noise addition is superfluous when combined to LoRA's regularization properties that we discuss in Section 5.

**Differential Privacy.** In Appendix H, we discuss the challenges of applying differential privacy (DP) in our federated setting and the alternative approaches we explored. DP is a well-established technique that provides formal guarantees to protect individual data from being inferred through the model's output. However, integrating DP into our setup requires significant modifications to our training pipeline that are beyond the scope of this work. Nevertheless, we show that applying gradient clipping without noise addition improves both accuracy and reduces memorization during fine-tuning, albeit without formal privacy guarantees. Furthermore, we show in Appendix I that injecting random Gaussian noise into the model weights does not improve the privacy–accuracy trade-off.

**Secure Aggregation.** While we observed lower memorization in FL compared to CL (see Section 4), locally trained models transmitted during FL may still expose participant data if not properly protected. In Appendix J, we present experiments using a secure aggregation protocol combining Fully Homomorphic Encryption (FHE) and Secure Multiparty Computation (SMPC). We find that this approach effectively mitigates privacy risks in the federated setting while introducing only negligible computational overhead.

## 5 Potential Theoretical Explanations

LoRA's memorization-mitigation effect remains largely empirical and lacks a theoretical foundation. Here, we discuss several works that present theoretical explanations as potential directions for future work. We

address additional theoretical work on the impact of FedAvg and similarities to $\delta$-compression operators in Appendix K.

**LoRA as regularization**. Biderman et al. (2024) presents LoRA as a competitive regularization method, and compares it to traditional regularization techniques. They find that LoRA retains more factual knowledge from its pretraining than fine-tuning with attention dropout or weight decay, and also keeps a better diversity of token generation than full fine-tuning. Indeed, our experiments show that LoRA is a competitive alternative to full fine-tuning for our medium-sized datasets, sometimes even slightly outperforming full fine-tuning. Furthermore, we found that combining LoRA with NEFTune, a noise-based regularization technique, does not lead to further improvement, suggesting a potential redundancy of regularization.

If we regard memorization as a function of duplication-induced overfitting, the preserved model accuracy of LoRA fine-tuning coupled with significantly lowered memorization may signal a reduction in benign-overfitting (Bartlett et al., 2020). That is, while full fine-tuning does not significantly alter model performance, the usage of full gradients may result in training overfitting without affecting generalization. As our gradients $\nabla W$ are low-rank, from a principal component analysis (PCA) perspective, excluding minor singular vectors in an update may reduce overfitting onto training data. Indeed, (Biderman et al., 2024) shows that full-finetuning learns updates with ranks $10 - 100\times$ larger than LoRA and Zeng & Lee (2024) formally and empirically show that any Transformer model with hidden dimension $d$ can be well-approximated with a LoRA rank of $d/2$. Thus, it is possible that LoRA reduces benign overfitting (Bartlett et al., 2020), which occurs when training data is overfitted without affecting performance. Notably, Tang et al. (2023a) prove that benign overfitting can preserve out-of-distribution generalization for overparameterized linear models if there is a strong correlation between the dominant eigenvectors/components of the source and target distributions. It is possible then that our LLMs are displaying this phenomenon: in both the centralized and FL settings, our fine-tuning datasets, while heterogeneous, contain aligned components due to their shared domain. LoRA may reduce benign overfitting by ignoring minor components, which only explain a minimal (and possibly noisy) portion of the data covariance.

**LoRA as DP-SGD.** Recent work by Malekmohammadi & Farnadi (2025) establishes a theoretical and empirical relationship between LoRA training and the DP-SGD algorithm. LoRA is shown to be approximately equivalent to fine-tuning adapters with noisy batch gradients, where the noise variance is a decreasing function of the LoRA rank. Indeed, we found that reducing the LoRA rank reduces unintended memorization, although at the cost of decreased performance, similarly to DP-SGD. Consequently, Malekmohammadi & Farnadi (2025) showed that LoRA provides additional robustness to membership inference attacks.

## 6  Conclusion and Limitations

In this work, we demonstrate that LoRA in FL is capable of reducing memorization of fine-tuning sensitive training data with little to no downstream performance cost. In particular, this effect is also observable in both centralized learning, and we analyze how memorization patterns differ between the two. Moreover, it is possible to further reduce memorization by combining LoRA with other strategies such as Goldfish loss or conventional privacy-preserving mechanisms such as Gaussian noising and gradient clipping.

Our study is limited to cross-silo settings with few clients, and further work is needed to analyze whether our findings generalize to large-scale cross-device settings. Additionally, further research on a theoretical explanation of our results is needed as well, and we discussed existing work and hypotheses for future directions in Section 5 and Appendix K such as reduction of benign overfitting (Bartlett et al., 2020), similarities between LoRA and DP-SGD (Malekmohammadi & Farnadi, 2025), and connections to $\delta$-compression operators (Karimireddy et al., 2019).

While we found significant memorization reduction in our study, LoRA and FL does not completely eliminate unintended memorization, even when combined with other privacy-enhancing techniques. As argued by Brown et al. (2022) in the context of LLM training, *the only truly privacy-preserving solution is to rely exclusively on data that is intended to be public* and LoRA should not be considered a panacea for unintended memorization, but rather a privacy-wise improvement over full fine-tuning.

**Broader Impact Statement**

This paper presents work whose goal is to advance the field of Machine Learning, especially enhancing privacy. Among the many potential societal consequences of our work, we specifically acknowledge that techniques mitigating unintended memorization can incidentally facilitate the concealment of unlawful use of copyrighted data by preventing its regurgitation post-training. However, we believe that the benefit of enhanced safeguards for confidential data protection combined with the current advances of other methods such as watermarking (Li et al., 2023a; Tang et al., 2023b; Cui et al., 2024b) can effectively mitigate this risk and provide stronger overall data protection.

**Acknowledgments**

This research is conducted as part of an *Innovation Project supported by Innosuisse*. The authors gratefully acknowledge financial support from Innosuisse under the Innovation Projects with Implementation Partner funding scheme. Additional support was provided by the European Union within the framework of the Phase IV AI Project.

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

## A    Further Related Work

**Differential privacy.** Classical $(\epsilon, \delta)$-differential privacy (DP) frameworks formally measure the privacy-preserving capacity of an algorithm by analyzing whether the probability of observing an output changes by $\epsilon$ when the underlying database excludes or includes a user record (Dwork et al., 2006). The application of this framework to generative language tasks, in general, has proven complicated due to the rigid definition of a user record (Brown et al., 2022; Jayaraman & Evans, 2019). When directly applying DP to prevent sensitive data reconstruction, it has been shown that a non-negligible compromise on privacy is required to maintain performance (Lukas et al., 2023). The conventional technique of adding Gaussian noise onto clipped gradients (Abadi et al., 2016) to boost privacy has also been shown to affect model outputs: the randomness of the noise alone can significantly alter the outputs of two equally-private models (Kulynych et al., 2023). McKenna et al. (2025) have recently found significant differences in the scaling laws of DP LLM training compared to models trained under traditional regime.

**Membership inference attacks** (MIA) rely on rigorous statistical principles to assess privacy risks in machine learning models. (Shokri et al., 2017) introduced an approach for determining whether a specific data point was part of a model's training dataset. These attacks exploit differences in model behavior on training versus non-training data, posing significant privacy concerns for sensitive information. Building on this, (Hongyan et al., 2024) extended these concepts to LLMs by incorporating contextual information. This study demonstrated that LLMs are particularly vulnerable to membership inference attacks, as they often retain verbatim information from their training datasets. The work highlighted the increased privacy risks associated with LLMs due to their scale and training dynamics.

**Secure Aggregations.** While the conventional FL ensures that raw data is not shared between participants during collective training, it does not address the risk of data leakage through model updates shared prior to aggregation. For example, in the honest-but-curious scenario, a server examines whether client data can be reconstructed (Huang et al., 2021). This vulnerability becomes particularly critical with LLMs, given their propensity for memorization. To address the privacy risks associated with local model exchanges in FL, (Truex et al., 2019) proposes a hybrid approach that combines differential privacy with secure multiparty computation (SMC). In this framework, local models are encrypted and remain hidden from other participants prior to aggregation, thereby mitigating privacy leakage risks associated with individual local models by focusing them on the aggregated model during each aggregation round. While this method has been explored for general machine learning applications, to the best of our knowledge, it has not yet been investigated in the context of large language models (LLMs).

**Medical applications.** Our emphasis on medical datasets is relevant: LLMs have been shown to regurgitate sensitive medical data in Lehman et al. (2021), though their work relies on an older BERT model. Mireshghallah et al. (2022) study the success of membership inference attacks on i2b2, though they also do not use any memorization metrics. Although federated learning has been studied and championed as an ideal paradigm for clinical settings (Xu et al., 2021; Nguyen et al., 2022; Antunes et al., 2022), there is a relative lack of literature in the context of clinical memorization.

## B    LoRA vs Full Fine-tuning Accuracy

To compare memorization between LoRA and full fine-tuning, it is essential that we compare settings that yield similar performance, as well as for this performance to be significantly higher than the base model. We use 3-shot in-context learning without any chain-of-thought reasoning and average the accuracy over three seeds. Figure 6 and 7 illustrate the resulting benchmark accuracy of the fine-tuned models we evaluate in this study, in federated learning and centralized learning respectively. We compare the base model to LoRA and full fine-tuning and find that the two fine-tuning methods reach a similar downstream accuracy that is significantly higher than the base model. A breakdown per benchmark in centralized learning is included in Table 2. Every fine-tuning yields a significant accuracy improvement over the pre-trained model except for Llama 3.1 8B as shown in Figure 7, which performance didn't improve with fine-tuning. Hyperparameter search either resulted in a significant performance drop (with high memorization) or kept the learning rate low enough that the accuracy stayed constant and thus showed no memorization. We hypothesize that part

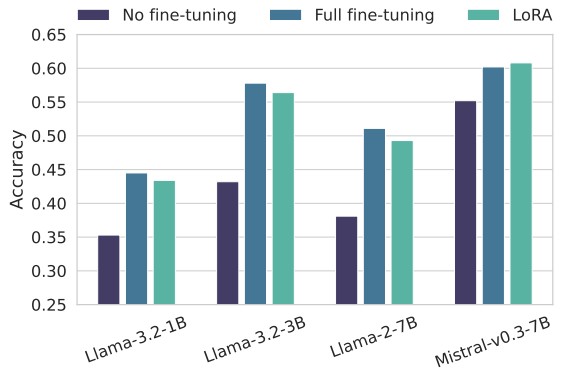
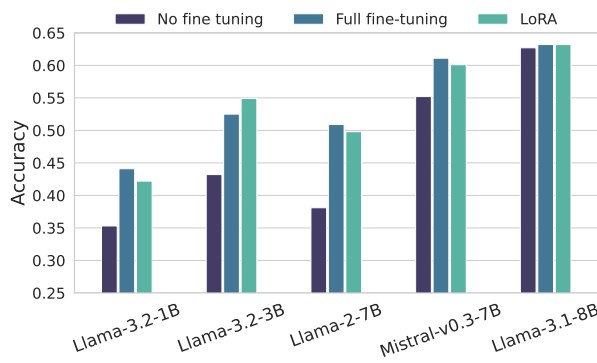

Figure 6: **Downstream accuracy in federated learning averaged across the 5 benchmarks.** LoRA yields relatively similar accuracy to full fine-tuning for several LLMs in a heterogeneous FL setting. We report the out-of-the-box accuracy of the pre-trained models as a control. A breakdown per round is included in Table 3.

Figure 7: **Centralized learning downstream accuracy averaged across the 5 benchmarks.** LoRA matches full fine-tuning accuracy on every model tested. A breakdown per benchmark is included in Table 2.

or all of our fine-tuning dataset has already been trained on during Llama 3.1 8B's pre-training phase, though the model showed no memorization of the canaries before fine-tuning. Accordingly, we exclude Llama 3.1 8B from subsequent experiments.

Table 2: **Downstream accuracy in centralized learning.** Best accuracy values are marked in **bold**.

| Model | Fine-tuning | MMLU-medical | PubMedQA | MedMCQA | MedQA | MedQA-4 | Average |
|---|---|---|---|---|---|---|---|
| Llama 3.2 1B | No fine-tuning | 0.353 | 0.363 | 0.49 | **0.329** | 0.275 | 0.308 |
| | Full | **0.456** | **0.616** | **0.431** | 0.322 | **0.379** | **0.441** |
| | LoRA | 0.447 | 0.594 | 0.397 | 0.312 | 0.362 | 0.422 |
| Llama 3.2 3B | No fine-tuning | 0.432 | 0.597 | 0.122 | 0.491 | 0.446 | 0.504 |
| | Full | 0.59 | 0.536 | **0.542** | **0.452** | **0.507** | 0.525 |
| | LoRA | **0.608** | **0.676** | 0.512 | 0.448 | 0.5 | **0.549** |
| Llama 2 7B | No fine-tuning | 0.381 | 0.426 | 0.452 | 0.380 | 0.292 | 0.353 |
| | Full | **0.562** | 0.596 | **0.516** | **0.395** | **0.478** | **0.509** |
| | LoRA | 0.560 | **0.726** | 0.448 | 0.353 | 0.405 | 0.498 |
| Mistral v0.3 7B | No fine-tuning | 0.552 | 0.635 | 0.7 | 0.483 | 0.438 | 0.503 |
| | Full | 0.659 | **0.758** | **0.588** | **0.499** | **0.551** | **0.611** |
| | LoRA | **0.667** | **0.758** | 0.572 | 0.467 | 0.54 | 0.601 |

In our federated learning experiments (Section 4), we compare memorization after 5 federated rounds. This ensures that we compare models that have been trained on the same tokens for the same number of times. Similarly, we do not make use of early stopping in centralized learning. Consequently, our methodology may not stop the fine-tuning at an optimal number of steps and some model may reach their best accuracy at different number of rounds. To measure this, we show in Table 3 the accuracy of federated fine-tuning per round. We find that except Llama 3.2 3B, all models reach their best performance in round 4 or 5, without any method reaching its best accuracy systematically earlier. For Llama 3.2 3B, LoRA reaches its highest accuracy at round 2 and then again at round 5. Since we compare memorization at the last round, LoRA memorization scores may be over-estimated due to overfitting, thus under-estimating how LoRA mitigates memorization compared to full fine-tuning.

Table 3: **Downstream accuracy per federated round**. We emphasize in **bold** the earliest round where models reach their best accuracy.

| Model | Fine-tuning | Accuracy per round | | | | |
|---|---|---|---|---|---|---|
| | | 1 | 2 | 3 | 4 | 5 |
| Llama 3.2 1B | Full | 0.425 | 0.438 | 0.444 | **0.445** | 0.445 |
| | LoRA | 0.415 | 0.422 | 0.430 | 0.432 | **0.434** |
| Llama 3.2 3B | Full | 0.541 | 0.561 | 0.554 | 0.573 | **0.578** |
| | LoRA | 0.557 | **0.564** | 0.559 | 0.563 | **0.564** |
| Llama 2 7B | Full | 0.468 | 0.488 | 0.482 | 0.495 | **0.511** |
| | LoRA | 0.475 | 0.490 | 0.482 | **0.494** | 0.493 |
| Mistral v0.3 7B | Full | 0.181 | 0.590 | 0.599 | **0.603** | 0.602 |
| | LoRA | 0.594 | 0.599 | 0.598 | 0.604 | **0.608** |

## C Generalization to other domains and larger models

To assess the robustness and broader applicability of our findings, we extend our evaluation beyond the medical domain and 7B-parameter LLMs. Specifically, we explore whether LoRA's memorization reduction in centralized learning persists in other high-risk domains such as law and finance, and to a 70B Llama model. These additional experiments demonstrate that our conclusions generalize across both tasks and model scales.

**Additional domains.** To evaluate generalizability beyond medicine, we fine-tuned models on Ai2's Multi-LexSum (Shen et al., 2022), a legal summarization dataset, and ConvFinQA (Cheng et al., 2024), a financial QA benchmark. *Multi-LexSum* contains long-form summaries of real-world civil rights lawsuits across multiple granularities. *ConvFinQA* is a conversational question-answering dataset derived from financial reports. It tests numerical reasoning and understanding in domain-specific contexts and includes sensitive financial information. These domains are highly sensitive to privacy risks, where even partial memorization can be problematic.

**Additional semantic measure.** We added BERTScore (Zhang et al., 2020) as an additional metric to better capture semantic similarity and subtle variations in memorized content. Following best practices, we use the *DeBERTa-xlarge-MNLI* model. We apply score rescaling, which adjusts for baseline similarity between unrelated sentence pairs. This technique improves the comparability and interpretability of reported scores across different models and datasets.

**Scaling up to 70B parameters.** We evaluated the LLaMA 3.1 70B Instruct model (Dubey et al., 2024) to test whether LoRA's memorization mitigation scales to larger models.

Table 4: **Law domain** memorization metrics. LoRA consistently lowers all metrics, including BLEU Score and BERT F1 score.

| Model | Method | BLEU | BERTScore |
|---|---|---|---|
| Llama 3.1 70B | Full FT | 0.55 | 0.55 |
| Llama 3.1 70B | LoRA | **0.17** | **0.32** |
| Llama 3.2 3B | Full FT | 0.29 | 0.42 |
| Llama 3.2 3B | LoRA | **0.06** | **0.17** |

**Results.** As shown in Table 4 for law and Table 5 for finance, LoRA consistently lowers BLEU and BERTScore compared to full fine-tuning, generalizing our findings in centralized learning on our medical dataset to other domains and measures. In particular, we also find that fine-tuning a 70B model with LoRA also yields lower memorization than full fine-tuning, indicating its continued effectiveness at scale. Memorization scores are

generally higher for the 70B model than for its 3B counterpart, except on the finance dataset, where we hypothesize that the lower memorization rate is due to suboptimal default hyperparameters.

Table 5: **Finance domain** memorization metrics. LoRA consistently lowers all metrics, including BLEU Score and BERT F1 score.

| Model | Method | BLEU | BERTScore |
|---|---|---|---|
| Llama 3.1 70B | Full FT | 0.55 | 0.48 |
| Llama 3.1 70B | LoRA | **0.50** | **0.45** |
| Llama 3.2 3B | Full FT | 0.51 | 0.56 |
| Llama 3.2 3B | LoRA | **0.11** | **0.12** |

**Lower duplication rate.** Previous duplication experiments relied on a duplication rate of 10. While we argue that such a rate is realistic in medical datasets given the prevalence of personal health information (PHI) in medical records, we further evaluate memorization with a duplication rate of 3 in Table 6. These results confirm that mitigation trends still holds with a lower duplication rate than the 10x duplication used in earlier sections, consistently with previous work (Wang & Li, 2025) in centralized learning.

Table 6: **Medical domain** memorization metrics for a large model and lower duplication rate. LoRA consistently lowers all metrics, including BLEU Score and BERT F1 score.

| Model | Dupl. | Method | BLEU | BERTScore |
|---|---|---|---|---|
| Llama 3.1 70B | None | Full FT | 0.170 | 0.23 |
| Llama 3.1 70B | None | LoRA | **0.100** | **0.18** |
| Llama 3.2 3B | None | Full FT | 0.030 | **0.11** |
| Llama 3.2 3B | None | LoRA | **0.010** | 0.12 |
| Llama 3.2 3B | 3x | Full FT | 0.060 | 0.20 |
| Llama 3.2 3B | 3x | LoRA | **0.004** | **0.14** |

# D  Training setup

All experiments were performed on a university-grade HPC furnished with nodes of 8 80GB A100 GPUs, with a Python 3.11.9 environment, PyTorch 2.4.0 and CUDA 12.1. Fine-tuning fit on a single GPU without parallelization for model sizes up to 8GB. Llama 3.1 70B was fine-tuned on 8 GPUs. A centralized fine-tuning lasts 3 GPU-hours in average. Including preliminary and failed experiments, centralized training amounts to around 350 GPU-hours. In federated experiments, each round corresponds to one epoch of each of the 3 datasets, in averaging lasting one GPU-hour, in addition to hyperparameters search amounting to roughly 20 GPU-hours per federated fine-tuning and totalling around 250 GPU-hours. Experiments on the LoRA rank, batch size, Goldfish loss, NEFTune, gradient clipping and Gaussian noise add 400 GPU-hours.

**Hyperparameters.** In centralized learning, we sweep the learning rate $\in \{1e-5, 5e-5, 1e-4, 5e-4\}$ for full fine-tuning experiments. For LoRA experiments, we search for learning rate values $\in \{5e-5, 1e-4, 5e-4, 1e-3\}$ as recommended by Biderman et al. (2024). In federated learning experiments, we sweep the learning rate on each dataset individually for one epoch, with the same set of values as in centralized learning.

For all experiments we fine-tune models with the AdamW optimizer (Loshchilov & Hutter, 2019) with default parameters ($\beta_1 = 0.9$, $\beta_2 = 0.999$, $\epsilon = 1e^{-8}$, weight decay of 0.01). We used a context length of 1024 and ensured that no text inputs were longer than the context length. We use a linear warmup of 100 steps with a cosine annealing schedule. Unless mentioned otherwise, we use a global batch size of 32 with gradient accumulation and gradient checkpointing. For LoRA fine-tuning with use a rank of 16, an alpha of 8, drop

out 0.05 and use adapters for all projection layers. We study the impact of the LoRA rank on memorization in Section 4.3.

# E   Datasets and pre-trained models

In this section, we describe the datasets and pre-trained models used in our experiments, including fine-tuning sources, generalization datasets, evaluation benchmarks, and licensing terms.

## E.1   Fine-tuning Datasets

In order to reproduce a plausible FL environment with non-IID data, we select 3 popular medical datasets with different types of QA.

1. *MedMCQA* (Pal et al., 2022) is composed of multiple-choice questions, containing almost 190k entrance exam questions (AIIMS & NEET PG). We fine-tune on the training split and leave aside validation data as a downstream evaluation benchmark.

2. *PubMedQA* (Jin et al., 2019) consists of Yes/No/Maybe questions created from PubMed abstracts. The dataset contains 1k expert-annotated (PQA-L) and 211k artificially generated QA instances (PQA-A). We include 500 questions from the train and validation sets of PQA-L and 50k questions of PQA-A.

3. *Medical Meadow flashcards* (Han et al., 2023) contains 39k questions created from Anki Medical Curriculum flashcards compiled by medical students. We include 10k instances for fine-tuning data.

## E.2   Medical Benchmarks

To measure the downstream performance of the fine-tuned models, we evaluate models on 4 medical benchmarks following existing methodology (Wu et al., 2023a; Singhal et al., 2023c;b; Chen et al., 2023): MedQA, PubMedQA, MedMCQA, and MMLU-Medical.

1. *MedQA's 4-option questions.* MedQA (Jin et al., 2020) consists of US Medical License Exam (USMLE) multiple-choice questions. The test set contains 1278 questions with both 4 and 5-option questions. Following Chen et al. (2023), we report each case separately, respectively MedQA-4 and MedQA.

2. *MedQA's 5-option questions.*

3. *PubMedQA*'s test set contains 500 expert-annotated questions. No artificially-generated questions are used during evaluation.

4. *MedMCQA*'s test set does not provide answer labels, therefore we rely on the validation set, containing 4183 instances, to benchmark downstream performance following Wu et al. (2023a) and Chen et al. (2023).

5. *MMLU-Medical.* MMLU (Hendrycks et al., 2021) is a collection of 4-option multiple-choice exam questions covering 57 subjects. We follow Chen et al. (2023) and select a subset of 9 subjects that are most relevant to medical and clinical knowledge: high school biology, college biology, college medicine, professional medicine, medical genetics, virology, clinical knowledge, nutrition, and anatomy, and group them into one medical-related benchmark: MMLU-Medical.

## E.3   Pre-trained models

To account for the effect of model size on memorization (Carlini et al., 2023b; Tirumala et al., 2022), we study pre-trained models ranging from 1B to 8B parameters: Llama 3.2 1B, Llama 3.2 3B, Llama 3 8B (Dubey et al., 2024), Llama 2 7B (Touvron et al., 2023), and Mistral 7B v0.3 (Jiang et al., 2023). We also include memorization-focused experiments with the Llama 3.1 70B Instruct model (Dubey et al., 2024) in Appendix C to evaluate how LoRA scales to larger-capacity models.

### E.4 Licenses and Terms of Use

We provide below the licenses and usage terms for all datasets and pretrained models used in our work.

#### E.4.1 Datasets

- **MedMCQA** (Wu et al., 2023a)
  Source: `https://huggingface.co/datasets/openlifescienceai/medmcqa`
  License: Apache License 2.0

- **PubMedQA** (Singhal et al., 2023b)
  Source: `https://huggingface.co/datasets/openlifescienceai/medmcqa`
  License: MIT License

- **Medical Meadow Flashcards** (Han et al., 2023)
  Source: `https://huggingface.co/medalpaca/medalpaca-7b`
  License: Creative Commons license family

- **i2b2 2014 De-identification Dataset** (Stubbs & Özlem Uzuner, 2015)
  Source: `https://www.i2b2.org/NLP/HeartDisease`
  License: Available under a Data Use Agreement from *Partners HealthCare*. Access requires registration and approval.

- **Multi-LexSum** (Shen et al., 2022)
  Source: `https://huggingface.co/datasets/allenai/multi_lexsum`
  License: Open Data Commons License Attribution family

- **ConvFinQA** (Cheng et al., 2024)
  Source: `https://github.com/czyssrs/ConvFinQA`
  License: MIT License

#### E.4.2 Pretrained Models

- **LLaMA 2** (Touvron et al., 2023)
  Source: `https://www.llama.com/llama-downloads`
  License: Llama 2 Community License Agreement

- **LLaMA 3** (Dubey et al., 2024)
  Source: `https://www.llama.com/llama-downloads`
  License: Llama 3.x Community License Agreement

- **Mistral 7B v0.3** (Jiang et al., 2023)
  Source: `https://huggingface.co/mistralai/Mistral-7B-v0.3`
  License: Apache License 2.0

## F Goldfish loss

The Goldfish loss (Hans et al., 2024) has been introduced recently as a memorization mitigating technique for pre-training language models via a new next-token training objective. The training procedure randomly excludes tokens from the loss computation in order to prevent verbatim reproduction of training sequences. While Goldfish loss has been designed for pre-training, we apply it to our fine-tuning and report values for various dropping frequencies $k$ and we use a hashing context width $h = 13$ following the authors' methodology (Hans et al., 2024). We evaluate the memorization and accuracy of Llama 3.2 3B fine-tuned with LoRA in combination with Goldfish loss. We also compare it to the same model fully fine-tuned with Goldfish loss only. Table 7 shows how combining Goldfish loss with LoRA mitigates memorization compared to a full fine-tuning. By contrasting memorization scores with control values, we can also note that the Goldfish loss is an effective memorization-mitigation technique.

Table 7: **Impact of Goldfish loss on BLEU Scores and accuracy in LoRA Fine-Tuning.** Llama 3.2 3B is fine-tuned with different dropping frequencies ($k$). Best accuracy is marked in **bold**.

| Goldfish $k$ | BLEU, no duplication | BLEU, 10x duplication | Accuracy |
|:---:|:---:|:---:|:---:|
| 2 | 0.0133 | 0.0216 | 0.514 |
| 3 | 0.0154 | 0.0426 | **0.549** |
| 4 | 0.0180 | 0.0543 | 0.534 |
| 5 | 0.0183 | 0.0815 | 0.540 |
| 10 | 0.0256 | 0.1494 | 0.538 |
| 100 | 0.0266 | 0.2852 | 0.537 |
| 1000 | 0.0256 | 0.3111 | 0.533 |
| 10000 | 0.0253 | 0.2944 | 0.545 |
| Control | 0.0245 | 0.2920 | 0.550 |

To assess the impact of LoRA in combination with Goldfish loss, we evaluated the memorization and accuracy of fine-tuning the same model using full fine-tuning. Table 8 presents the memorization scores and accuracy of the model fine-tuned with Goldfish loss alone, without LoRA. Our results indicate that while Goldfish loss reduces memorization, it does not achieve the same level of reduction as the combination with LoRA, especially when duplication occurs in the fine-tuning data. In summary, combining LoRA with Goldfish loss allows a privacy-utility tradeoff that cannot be achieved using Goldfish loss alone.

Table 8: **Impact of Goldfish loss on BLEU Scores and accuracy in full fine-tuning.** The BLEU scores and the accuracy of Llama 3.2 3B is reported for full fine-tuning across different dropping frequencies ($k$). Best accuracy is marked in **bold**.

| Goldfish $k$ | BLEU, no duplication | BLEU, 10x duplication | Accuracy |
|:---:|:---:|:---:|:---:|
| 2 | 0.0146 | 0.0340 | 0.517 |
| 3 | 0.0243 | 0.0679 | 0.513 |
| 4 | 0.0282 | 0.1148 | 0.524 |
| 5 | 0.0310 | 0.1568 | 0.521 |
| 10 | 0.0342 | 0.3006 | **0.545** |
| 100 | 0.0399 | 0.5821 | 0.534 |
| 1000 | 0.0425 | 0.6235 | 0.527 |
| 10000 | 0.0407 | 0.6235 | 0.516 |
| Control | 0.0417 | 0.6235 | 0.538 |

## G NEFTune

NEFTune is a regularization technique consisting in adding random noise to the embedding vectors to improve instruction fine-tuning. While not introduced as a privacy-preserving technique per se, we hypothesize that a fine-tuning regularization such as NEFTune may also reduce unintended memorization.

We display results after applying NEFTune with noise value $\alpha \in \{5, 10, 15, 30, 45\}$. We find that adding noise does not improve accuracy when applied to our domain adaptation fine-tuning. Secondly, increasing the noise does not yield better privacy, at least not until we set alpha to 45, which is greater than alpha values reported by the original work (5, 10, and 15).

## H Differential Privacy

$(\epsilon, \delta)$-Differential privacy (DP) provides formal guarantees that an individual's data cannot be inferred from a model's output, by quantifying the model's sensitivity to changes in input data. Following Li et al. (2021) and Liu et al. (2024), we define sensitivity as the maximum change in model output resulting from the inclusion or removal of a single data point in the training dataset (record-level DP).

Implementing differential privacy (DP) requires modifications to the fine-tuning pipeline to measure the sensitivity of the fine-tuning process. However, the use of stochastic gradient descent (SGD), required for DP-SGD, poses challenges when fine-tuning the Llama 3.2 3B model. Despite extensive hyperparameter

Table 9: **NEFTune impact on the BLEU score and accuracy when combined with LoRA.** We analyze LoRA fine-tuning with Llama 3.2 3B and different noise scaling factors $\alpha$.

| $\alpha$ | No duplication | 10x duplication | Accuracy |
|---|---|---|---|
| Control | 0.0276 | 0.4170 | 0.562 |
| 5 | 0.0284 | 0.4525 | 0.560 |
| 10 | 0.0300 | 0.4506 | 0.518 |
| 15 | 0.0284 | 0.4525 | 0.544 |
| 30 | 0.0282 | 0.4377 | 0.548 |
| 45 | 0.0248 | 0.3599 | 0.518 |
| 60 | 0.0227 | 0.2759 | 0.501 |
| 100 | 0.0183 | 0.1006 | 0.391 |

tuning, particularly of the learning rate, SGD consistently underperforms compared to Adam-based optimizers. As shown in Figure 8, Paged AdamW achieves substantially faster and deeper loss reduction than SGD during fine-tuning, highlighting the difficulty of maintaining optimization efficiency while measuring the sensitivity in large-scale models.

**Gradient clipping.** A key technique in differential privacy (DP) is gradient clipping, which constrains the magnitude of gradient updates and enables the measurement of model sensitivity during fine-tuning. In our experiments, we find that gradient clipping alone can mitigate memorization, even though it does not provide formal privacy guarantees. Using a clipping value of 0.0001 substantially reduced memorization and improved accuracy compared to the default value of 1.0. Table 10 reports the impact of different clipping values on BLEU score and accuracy during fine-tuning of the Llama 3.2 3B model. These results highlight gradient clipping as an effective privacy-enhancing mechanism in its own right, even without the addition of noise.

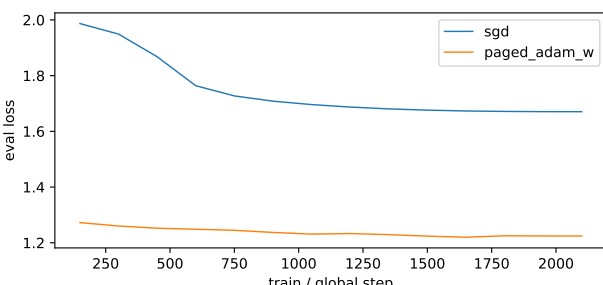

Figure 8: **Loss reduction comparison between optimizers.** The plot compares loss reduction during the fine-tuning of Llama 3.2 3B using different optimizers: SGD (blue) and Paged AdamW (orange).

# I    Post-fine-tuning Gaussian noise injection

This section provides details and results of the injection of noise into the weights of a model after fine-tuning. Specifically, the noise is sampled from a Gaussian distribution $\mathcal{N}(\mu, \sigma^2)$, where the mean $\mu$ is set to 0, and $\sigma^2$ is the variance that determines the noise's magnitude. Unlike the DP Gaussian mechanism, this approach does not provide formal privacy guarantees. However, it offers a practical and computationally light method to mitigate the memorization of sensitive information, as it does not require additional fine-tuning and can be directly applied to previously fine-tuned LLMs. Additionally, measuring the performance of this method can illustrate how other noise mechanisms similar to those used in DP might affect accuracy and privacy metrics.

In Table 11, we evaluate its effect under various noise magnitudes, along with the corresponding impact on model accuracy. We applied Gaussian noise to the LoRA weights of a fine-tuned Llama 3.2 3B model, as evaluated in earlier sections. We then compared the model's BLEU score and accuracy across different noise magnitudes.

Table 10: **Gradient clipping impact on the BLEU score and accuracy.** The BLEU score and the accuracy of Llama 3.2 3B is reported for LoRA fine-tuning. Best accuracy is marked in **bold**.

| Clipping Value | No duplication | 10x duplication | Accuracy |
|---|---|---|---|
| $1.0 \times 10^{0}$ (default) | 0.0266 | 0.4235 | 0.520 |
| $5.0 \times 10^{-1}$ | 0.0235 | 0.4235 | **0.541** |
| $1.0 \times 10^{-1}$ | 0.0229 | 0.4031 | 0.530 |
| $5.0 \times 10^{-2}$ | 0.0243 | 0.3827 | 0.534 |
| $1.0 \times 10^{-2}$ | 0.0227 | 0.3914 | 0.506 |
| $5.0 \times 10^{-3}$ | 0.0245 | 0.3914 | 0.531 |
| $1.0 \times 10^{-3}$ | 0.0250 | 0.3352 | 0.519 |
| $5.0 \times 10^{-4}$ | 0.0203 | 0.2914 | 0.528 |
| $1.0 \times 10^{-4}$ | 0.0185 | 0.0926 | 0.536 |
| $5.0 \times 10^{-5}$ | 0.0151 | 0.0438 | 0.506 |
| $1.0 \times 10^{-5}$ | 0.0086 | 0.0099 | 0.491 |
| $5.0 \times 10^{-6}$ | 0.0065 | 0.0080 | 0.449 |
| $1.0 \times 10^{-6}$ | 0.0026 | 0.0012 | 0.460 |
| $5.0 \times 10^{-7}$ | 0.0026 | 0.0012 | 0.392 |
| $1.0 \times 10^{-7}$ | 0.0026 | 0.0012 | 0.377 |

Table 11: **Impact of noise addition on BLEU score and accuracy.** Llama 3.2 3B is fine-tuned with LoRA across various noise magnitudes ($\sigma$)

| Noise Scale ($\sigma$) | BLEU, no Duplication | BLEU, 10x Duplication | Accuracy |
|---|---|---|---|
| 0 (no noise) | 0.0206 | 0.3012 | 0.553 |
| 0.001 | 0.0211 | 0.3049 | 0.552 |
| 0.01 | 0.0206 | 0.2877 | 0.551 |
| 0.02 | 0.0143 | 0.0994 | 0.541 |
| 0.03 | 0.0083 | 0.0111 | 0.511 |
| 0.04 | 0.0013 | 0.0006 | 0.384 |
| 0.05 | 0.0000 | 0.0000 | 0.110 |

We observe that the accuracy remains unaffected up to a certain noise level ($\sigma = 0.01$) and even shows slight improvement. However, beyond this threshold, accuracy decreases and reduction in memorization similarly follows, appearing to correlate with this decrease. These observations suggest that this mechanism effectively reduces excessive memorization in models that have overfitted onto their training data. Therefore, this approach offers an alternative to early stopping for controlling memorization which can be applied post fine-tuning. Figure 9 compares the privacy and utility of Llama 3.2 3B subject to post-fine-tuning gaussian noise injection with the evolution of the model fine-tuned with LoRA accross iterations. The noisy model, represented by red dots, has been fine-tuned for 2100 iterations before injecting the gaussian noise. Gaussian noise injection of standard deviations of $\sigma = 0.2$ and $\sigma = 0.3$ have been reported in the plot.

### I.1 Privacy-Utility tradeoff with Gaussian noise injection

Figure 9 presents a dot plot comparing the privacy-utility tradeoffs of Llama 3.2 3B when fine-tuned with LoRA versus when Gaussian noise is injected after fine-tuning with LoRA. The results indicate that Gaussian noise injection does not enhance the privacy-utility tradeoff compared to fine-tuning with LoRA.

## J  Secure Aggregations

Secure aggregation ensures that sensitive data remains protected by preventing the aggregator from decrypting any individual model updates. In a federated learning setup, only the aggregated global model is accessible to participants, meaning that memorization within locally trained models before aggregation does not pose a privacy risk. We also evaluate the runtime performance of combining secure aggregation with LoRA in a federated learning setting, demonstrating that this added protection incurs minimal computational overhead.

**Performance.** To evaluate the performance impact of secure aggregation, we use Lattigo, an open-source library that enables secure protocols based on multiparty homomorphic encryption Lattigo v6; Mouchet et al.

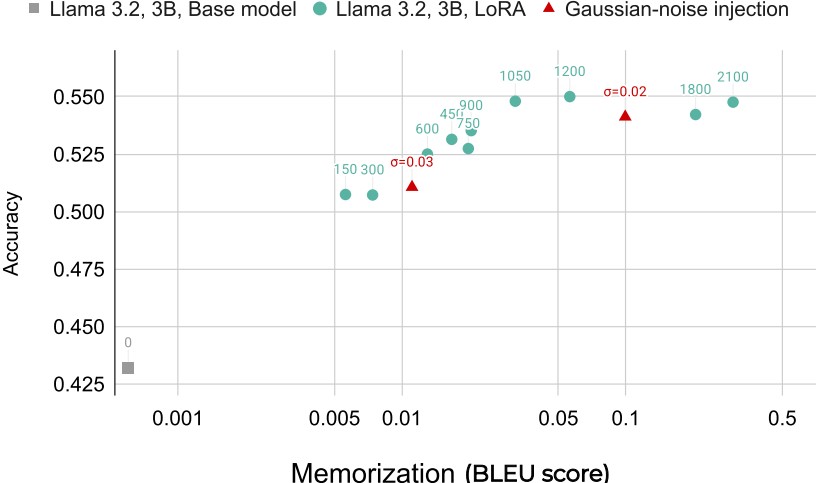

Figure 9: **Privacy-Utility tradeoff with post-fine-tuning gaussian noise injection.** Accuracy and memorization (BLEU score with 10x document duplication) tradeoff of Llama 3.2 3B subject to post-fine-tuning gaussian noise injection with standard deviation. Values above the dots correspond to the number of iterations for LoRA fine-tuning evolution, and the standard deviation of injected noise for noisy models.

(2020). Specifically, it implements the CKKS scheme, which allows efficient encrypted computations on real-valued data, making it ideal for the secure aggregation of the LoRA models trained by the clients/participants. In our experiments, we consider 3 clients and configure CKKS parameters to enable 32-bit precision. Since our LoRA models are trained with 16-bit precision, this ensures that **secure aggregation does not introduce any accuracy loss** compared to standard aggregation in plaintext.

**Time overhead.** Secure aggregation introduces a time overhead due to encryption, homomorphic operations, and collective decryption. The duration of encrypted aggregation is influenced by the number of weights being aggregated, specifically the number of LoRA weights. In our experiments with Llama 3.2 3B, **a LoRA update contains 24,772,608 parameters, representing approximately 0.77% of the full model's parameters**. In Table 12, we report the aggregation times for vectors of varying sizes, corresponding to the number of LoRA weights. Aggregating three vectors of the size of our LoRA takes 11.33 seconds, which is negligible compared to the time required for local fine-tuning at each round.

Table 12: **Execution Time of the Secure Aggregation Protocol.** The protocol aggregates three equal-sized encrypted vectors for varying sizes.

| Aggregation Length | Time Taken |
|---|---|
| $10^1$ | 12.16ms |
| $10^2$ | 11.61ms |
| $10^3$ | 11.32ms |
| $10^4$ | 17.29ms |
| $10^5$ | 58.91ms |
| $10^6$ | 474.46ms |
| $10^7$ | 4.37s |
| $2.48 \times 10^7$ (LoRA size) | **11.33s** |
| $10^8$ | 68.24s |

## K   Why Does LoRA Reduce Memorization?

We continue here the theoretical discussion started in Section 5. Our experimental evaluation demonstrates that LoRA reduces memorization in both centralized and FL settings, which naturally raises the question:

*why does this happen?* We argue that the mechanisms by which FedAvg and LoRA mitigate memorization should be considered independently. Carlini et al. (2023a) empirically establish a log-linear relationship between canary duplication and memorization, thus we frame our discussion of memorization in the context of overfitting. How and why in-distribution, non-duplicated sequences can still be regurgitated (Carlini et al., 2019) is a question that we leave to future work.

**Federated learning.** While it is known that FedAvg can reduce memorization for simpler LSTM-based next-word predictors (NWPs) (Ramaswamy et al., 2020; Thakkar et al., 2021), we hope that our verification of this phenomenon for LLMs on longer canaries can encourage formal investigation. Nevertheless, we note the following: in the IID FedAvg setting with identical hyperparameter settings (same number of local updates, learning rate, and initialization) the expected value of the $d$-sample stochastic gradient over $N$ clients, $\frac{1}{N}\frac{1}{d}\sum_{i=1}^{k} f_k(\theta, x_i \sim D_k)$ in Equation 1 can resemble a single stochastic gradient in a centralized setting taken over a single large batch of size $Nk$ since $f_k$ and $D_k$ are homogeneous. Thus, Thakkar et al. (2021) observe more memorization in IID settings with larger batch sizes. The non-IID setting is significantly more complex: the optimization problem and associated loss landscape of Equation 1 differs from the centralized problem. We observe in Figure 2 that non-IID FL significantly reduces memorization, which Thakkar et al. (2021) also observe for their NWPs. While they do not fine-tune their learning rates to eliminate this as a confounding variable, we do[2], thus suggesting that FedAvg itself is a memorization-reducing mechanism.

**$\delta$-compressors.** Specific to FL, an alternative hypothesis is that the low-rank approximation of $\Delta W$ resembles a $\delta$-compression operator (Karimireddy et al., 2019), i.e., $||\mathtt{LORA}(\Delta W) - \Delta W||^2 \leq (1 - \delta)||\Delta W||^2$, and that low-$\delta$ compressors reduce memorization. Low-bias compressors, such as certain randomized projections (Dorfman et al., 2023; Rabbani et al., 2021; Ivkin et al., 2019) and other low-rank approximations (Makkuva et al., 2023) have been shown to preserve model performance in non-IID distributed settings. While the effects of these other operators on memorization has not been extensively studied, the efficacy of gradient clipping in lowering memorization while maintaining accuracy (Table 10) lends further credence to this hypothesis. Clipping is a low-bias compressor for heavy-tailed gradients, which is observed for general SGD (Mireshghallah et al., 2022) and LLM fine-tuning (Kenton & Toutanova, 2019). Further exploration of $\delta$-compressors such as sketches, signSGD (Bernstein et al., 2018), QLoRA (Dettmers et al., 2024), and U-Clip (Elesedy & Hutter, 2023) is warranted.

---

[2]While it is possible that performing centralized learning in a curriculum-style manner with heterogeneous learning rates over training data can reduce memorization, given the small performance gap against non-IID FL, it is highly unlikely that this alone can improve its significantly worse memorization scores.

