# OpenReview forum: "Mitigating Unintended Memorization with LoRA in Federated Learning for LLMs"
_TMLR — Accepted by TMLR_

### Review · Reviewer_86LA · 2025-10-29

**Summary Of Contributions:**

This paper empirically studies the effect of federated learning on unintended memorization when using LoRA.
The authors provide empirical evidence that Federated LoRA memorizes less than both Federated Fine-Tuning and Centralized LoRA. Experiments are conducted on realistic datasets with large, practical LLMs such as LLaMA-2, LLaMA-3, and Mistral. The authors also explore combining LoRA with other privacy-preserving techniques.

The topic is relevant and timely. However, the findings are not particularly surprising, as prior work has already shown that, in the centralized setting, LoRA tends to memorize less than full fine-tuning (as it can be viewed as a form of noisy gradient descent; see Malekmohammadi & Farnadi, 2025), and that federated learning typically exhibits less memorization than centralized training (Thakkar et al., 2021). Based on the discussion in Appendix L, the current study does not seem to bring substantial new insights beyond these known results.

Weaknesses
- Weak discussion of empirical results.
The main discussion of the findings is relegated to the appendix. This part should be integrated into the main text and clearly linked to the empirical observations.
- Unclear organization of experimental results.
The main message—that federated learning improves privacy compared with the centralized setting, and even more so when combined with LoRA—is not clearly conveyed. Some key results (e.g., Section 4.1 and parts of Section 4.2) are presented only for the centralized case, making it difficult to verify whether the same conclusions hold in the federated setting.
- Questions regarding Figure 3:
  - For LLaMA-2-7B and Mistral, federated learning occasionally shows higher memorization than the centralized setup. It is not convincing that this is solely due to differing learning rates.
  - Why is the centralized curve stable across rounds? Some clarification is needed.

- Figure 6 should appear in the main text.
Its content is essential for the paper’s claims. It would also be helpful to include both centralized and federated scenarios in the same figure. Additionally, I was surprised to observe that, in the non-IID case, federated learning outperforms centralized learning (Tables 2 and 3), which contradicts theoretical expectations. Moreover, LoRA showing better performance than full fine-tuning (Figure 6) is counterintuitive, as LoRA is theoretically akin to DP-SGD and thus expected to yield a slight performance trade-off.
- Experimental design could be strengthened.
The paper cites Thakkar et al. (2021) in Appendix L, where the experiments were carefully designed to isolate different contributing factors to memorization. A similar level of rigor in experimental design would have provided deeper insights here.

**Audience:**

Yes

**Audience Explanation:**

Yes, especially those interested in privacy and efficient fine-tuning methods.

**Broader Impact Concerns:**

The paper raises no immediate ethical or societal concerns

**Claims And Evidence:**

No

**Claims Explanation:**

The claims made in the submission are partially supported by empirical evidence, but the support is neither fully convincing nor clearly presented. While the experiments confirm previously known trends—such as LoRA and federated learning both reducing memorization—the empirical analysis lacks depth, and the discussion is mostly deferred to the appendix. Some results (e.g., Figures 3 and 6) even appear inconsistent or counterintuitive and are not adequately explained. Overall, the evidence provided does not convincingly substantiate the paper’s claims or bring new insights beyond prior work.

**Requested Changes:**

- Strengthen the discussion of empirical findings. Move the key points from Appendix L into the main text and clearly connect them with the presented results. (Critical)

- Reorganize the experimental results to ensure both centralized and federated scenarios are consistently compared. Emphasize the main message that federated learning and LoRA together improve privacy. (Critical)

- Improve experimental design by incorporating analyses that isolate contributing factors to memorization, following the methodological rigor of Thakkar et al. (2021). (Critical)

- Enhance overall clarity and organization, ensuring that the narrative highlights the paper’s contributions and situates them in the context of existing literature.

---

> ### Author Response · Authors · 2025-11-23
> **Response to Reviewer 86LA (1/2)**
>
> We thank the reviewer for the thorough assessment of our work and the valuable feedback.
>
> > Reorganize the experimental results to ensure both centralized and federated scenarios are consistently compared. Emphasize the main message that federated learning and LoRA together improve privacy. (Critical)
>
> We reorganized our Results Section to incorporate our main findings as such:
>
> 1. We introduce our experiments in FL and CL side-to-side to facilitate comparisons and moved the CL results from the Appendix into the main body.
> 2. We move and discuss the privacy-utility tradeoff directly in the main body.
> 3. Finally, we clearly state our main findings in each section and emphasize them with bold type
>
> > Move the key points from Appendix L into the main text and clearly connect them with the presented results. (Critical)
>
> We move the main points from Appendix L into a new Section “Potential Theoretical Explanations” and link each theoretical point to our empirical findings.
>
> > Improve experimental design by incorporating analyses that isolate contributing factors to memorization, following the methodological rigor of Thakkar et al. (2021). (Critical)
>
> We fully agree with the reviewer that our analysis would benefit from a more systematic evaluation of each factor in isolation. Nevertheless, our experimental design already isolates many factors: memorization metrics, data duplication, prompt lengths, CL vs FL and of course LoRA vs full-fine-tuning. Thakkar et al.‘s combinatorial design is facilitated by the small size of their model, composed of only 1.3M parameters, while the multi-billion parameter scale of recent models makes this approach computationally prohibitive. Still, their work has inspired us to extend our analysis to realistic canaries instead of random sequences and their findings informed our decision of using a data heterogeneous setting.
>
> Enhance overall clarity and organization, ensuring that the narrative highlights the paper’s contributions and situates them in the context of existing literature.
>
> We fully revised our Introduction and Motivation in order to:
>
> *  better frame the problem we are addressing (LLM memorization in FL),
> * discuss the literature gap (LoRA and FL has only been studied under DP theoretical bounds, yet past work successfully performed extraction attacks on DP-trained LLMs [1])
> * and present how our contributions fill this gap (we empirically measure memorization from combining LoRA and FL in a variety of settings).
>
> [1] “Analyzing leakage of personally identifiable information in language models”, Lukas et al., IEEE, 2023
>
> > Questions regarding Figure 3 [Figure 4 in the revised version]: For LLaMA-2-7B and Mistral, federated learning occasionally shows higher memorization than the centralized setup. It is not convincing that this is solely due to differing learning rates.
>
> Indeed, in some cases the FL curves in Figure 3 (Figure 4 now) shows higher memorization than the centralized baseline. This is partly due to the comparison setup, where the centralized model is evaluated at a single checkpoint after 3 epochs, while the FL curve reflects memorization across many rounds, which can amount to more overall training and lead to additional overfitting. We also note a few instances where FL shows higher memorization even before 3 federated rounds (i.e., before reaching the amount of training used for the centralized baseline). Our hypothesis is that this comes from the non-IID client data, where local training on domain-specific distributions can increase memorization early on, and the aggregation step may not reduce it as effectively as centralized training. We clarified this point in section 4.1 "LoRA in FL vs CL".
>
> > Questions regarding Figure 3 [Figure 4 in the revised version]: Why is the centralized curve stable across rounds? Some clarification is needed.
>
> To address the concern about the centralized curve appearing “stable across rounds,” we updated Figure 4 (formerly Figure 3) so that it compares only LoRA and full fine-tuning within the federated setting, without plotting the centralized baseline across rounds. This removes the misleading impression that the centralized model remains stable while FL memorization evolves during training. The CL–FL comparison is now more clearly and fairly presented by examining Figures 2 and 3 together, where the number of epochs (CL) and federated rounds (FL) are fixed in both settings.
>
> (continued)

---

> > ### Author Response · Authors · 2025-11-23
> > **Continued Response to Reviewer 86LA (2/2)**
> >
> > (continued)
> >
> > > Figure 6 should appear in the main text. Its content is essential for the paper’s claims. It would also be helpful to include both centralized and federated scenarios in the same figure. Additionally, I was surprised to observe that, in the non-IID case, federated learning outperforms centralized learning (Tables 2 and 3), which contradicts theoretical expectations. Moreover, LoRA showing better performance than full fine-tuning (Figure 6) is counterintuitive, as LoRA is theoretically akin to DP-SGD and thus expected to yield a slight performance trade-off.
> >
> > We have moved the original Figure 6, along with the section “The privacy–utility tradeoff”, into the main paper (now Figure 5), as recommended. Regarding the suggestion to combine centralized and federated results into a single figure, we agree that having a comparable figure for the federated setting would be valuable. However, producing such a figure would have required additional experimentation and aligned evaluation procedures that were not feasible within the revision timeline. We therefore leave the construction of a unified CL–FL visualization for future work, and instead ensured that Figures 2, 3, and Tables 2, 3 follow a consistent presentation to facilitate comparison across settings.
> >
> > Concerning the surprising result that FL outperforms CL in the non-IID case (Tables 2 and 3), we agree that this is unexpected. We hypothesize that, in this heterogeneous setting, the FL aggregation steps may act as an implicit regularization mechanism as argued by Biderman et al., 2024 [2], blending domain-specific updates from different clients. In contrast, the centralized model—trained on the fully merged non-IID dataset appears more prone to overfitting, which prevents it from reaching the performance achieved by FL under the same conditions.
> >
> > Finally, regarding LoRA outperforming full fine-tuning, we believe this is primarily due to LoRA’s stronger resistance to overfitting. Full fine-tuning showed clear signs of overfitting after several hundred steps, which likely limited its maximum achievable accuracy. Additional regularization techniques could have improved its performance, but these were not explored in the present work. As a secondary detail, full fine-tuning metrics were logged every 150 iterations, which may have missed its exact peak accuracy, though this effect appears minor compared to overfitting itself. We clarify these points in Section 4.2 of the revised manuscript.
> >
> > [2] “LoRA Learns Less and Forgets Less”, Biderman et al., TMLR, 2024

---

> > > ### Comment · Reviewer_86LA · 2026-01-07
> > >
> > > The authors have addressed my main concerns.

---

### Review · Reviewer_Da1W · 2025-11-02

**Summary Of Contributions:**

This paper makes the following key contributions:
1. Empirical demonstration that using Low-Rank Adaptation (LoRA) for fine-tuning Large Language Models (LLMs) in a Federated Learning (FL) setting significantly reduces unintended memorization of sensitive training data (by up to 10x), with little to no cost to downstream task performance.
2. The study is extensive, validating the core finding across multiple model families (Llama-2, Llama-3, Mistral), model scales (1B to 70B parameters), and high-risk domains (medicine, law, finance).
3. The work provides a detailed comparison of memorization patterns between Federated and Centralized Learning (CL), confirming that FL itself reduces memorization and showing that LoRA's mitigating effect is present in both settings.
4. The paper experimentally shows that LoRA can be effectively combined with other privacy-enhancing methods. It finds synergistic effects with Goldfish loss and gradient clipping, providing a pathway for building stronger, multi-layered privacy defenses.


Key Strengths:
+ The work addresses a critical, real-world problem of data leakage in privacy-sensitive applications of LLMs.
+ The experimental setup uses established models, memorization metrics (exact match, BLEU) and canaries on sensitive, real-world data (i2b2).
+ The findings for practitioners are clear: using LoRA is a simple and efficient way to gain a significant privacy improvement during fine-tuning.

Key Weaknesses:
- The introduction's logic flow is confusing and does not effectively set up the research gap. It lacks a clear, concise explanation of how FL is implemented for LLMs in this work.
 -   The claim that LoRA's privacy effects are unstudied in FL is inaccurate and ignores relevant prior work. E.g., Sun et al. has also explored the privacy-preserving FL via LoRA.
- The paper is predominantly empirical without any theoretical analysis. While Appendix L offers several compelling hypotheses (e.g., reduction of "benign overfitting," connection to differential privacy via low-rank compression), these are not proven and remain speculative.
- The FL setup uses only three clients. A deeper investigation into how the number of clients, data heterogeneity, and aggregation frequency affect memorization with LoRA would strengthen the FL-specific claims.

**Audience:**

Yes

**Audience Explanation:**

The findings are highly relevant to a significant portion of the TMLR audience. Key interested groups would include FL and LLM Practitioners, and Privacy and Security Researchers.

**Broader Impact Concerns:**

The most significant concern is the potential for practitioners to misinterpret the findings as offering comprehensive privacy protection. While the paper demonstrates a reduction in memorization, it does not eliminate it. Nonetheless, organizations in sensitive domains such as healthcare may adopt this technique with unwarranted confidence. Without explicit warnings, this could result in privacy breaches, as determined adversaries might still be able to extract sensitive information.

**Claims And Evidence:**

No

**Claims Explanation:**

While the core experimental evidence for LoRA reducing memorization is extensive and appears sound, the paper's clarity and contextualization are lacking, which undermines the overall conviction.

* The introduction's motivation is confusing, and the FL methodology is unclear, making it difficult for the reader to fully grasp the experimental setup.
* The evidence is presented in a vacuum regarding related work. By not engaging with papers like Sun et al., it's unclear how this work specifically advances the state-of-the-art beyond what has already been explored in "privacy-preserving FL via LoRA."

**Requested Changes:**

## Critical Changes (Required for Acceptance):

- **Revise the Introduction and Motivation**: The introduction must be restructured to present a clear, logical narrative. It should: a) clearly state the problem (LLMs memorize data in FL), b) identify the specific gap (while LoRA is known for efficiency, its specific effect on memorization in FL is not well-understood, especially compared to full fine-tuning and across diverse settings), and c) precisely state the paper's contributions in light of this gap. The current motivational flow is confusing and must be improved.

- **Acknowledge and Differentiate from Prior Work**: The statement "the effects of LoRA on privacy and memorization haven’t been formally studied in federated learning settings" must be corrected. The authors must cite and discuss relevant works like Sun et al. ("Improving lora in privacy-preserving federated learning") and clearly articulate how their work differs—for instance, by focusing explicitly on memorization metrics (exact match, BLEU) rather than differential privacy bounds, or by providing a much broader evaluation across model families and sizes.

## Strengthening Changes (Recommended to Improve the Paper):

- **Address the Limitation of a 3-Client Setup**: The paper must include a explicit discussion of this limitation. The authors should justify why this small-N setup was chosen (e.g., computational constraints, simulating a cross-silo scenario with a few hospitals) and acknowledge that the findings may not generalize to large-scale cross-device FL with hundreds or thousands of clients. This should be added to the "Limitations" section.

- **Refine the Theoretical Discussion**: The hypotheses in Appendix L should be framed more clearly as speculative and be positioned as directions for future work.

---

> ### Author Response · Authors · 2025-11-23
> **Response to Reviewer Da1W**
>
> We are thankful for the constructive assessment of our work and agree with all of your suggestions.
>
> > Revise the Introduction and Motivation: The introduction must be restructured to present a clear, logical narrative. It should: a) clearly state the problem (LLMs memorize data in FL), b) identify the specific gap (while LoRA is known for efficiency, its specific effect on memorization in FL is not well-understood, especially compared to full fine-tuning and across diverse settings), and c) precisely state the paper's contributions in light of this gap. The current motivational flow is confusing and must be improved.
>
> We agree with the reviewer, and fully revise our Introduction Section accordingly. The Introduction and Motivation is now composed of 4 paragraphs, which we articulated as follows:
>
> 1. Background: FL and LLMs are being used for data-sensitive tasks
> 2. Problem 1: We can expect FL-trained LLMs to memorize data and the impact of FL on memorization is unclear for large-scale LLMs.
> 3. Problem 2: LoRA is getting widely adopted in FL, yet we don’t understand its impact on privacy either. We discuss previous work by Sun et al. and additionally by Liu et al. [1] on LoRA + DP.
>     We identify the literature gap: unintended memorization hasn’t been empirically evaluated when combining LoRA and FL but only studied under DP theoretical bounds.
> 4. Our contributions: We extensively measure and compare empirical memorization across a variety of settings. We explicitly articulate how our work complements Thakkar et al, Sun et al. and Liu et al. and argue for the necessity of empirical evaluations in addition to DP theoretical guarantees.
>
> > Acknowledge and Differentiate from Prior Work: The statement "the effects of LoRA on privacy and memorization haven’t been formally studied in federated learning settings" must be corrected. The authors must cite and discuss relevant works like Sun et al. ("Improving lora in privacy-preserving federated learning") and clearly articulate how their work differs—for instance, by focusing explicitly on memorization metrics (exact match, BLEU) rather than differential privacy bounds, or by providing a much broader evaluation across model families and sizes.
>
> Thank you for pointing this gap in our introduction. Our revised introduction now explicitly discusses Sun et al‘s and Liu et al.‘s previous work on combining LoRA with DP-SGD and differentiate our work by focusing on its empirical nature. The last paragraph further motives why an empirical evaluation is necessary to complement work relying exclusively on DP theoretical bounds, given that prior work successfully performed training data extraction attacks on DP-trained LLMs [2].
>
> > Address the Limitation of a 3-Client Setup: The paper must include a explicit discussion of this limitation. The authors should justify why this small-N setup was chosen (e.g., computational constraints, simulating a cross-silo scenario with a few hospitals) and acknowledge that the findings may not generalize to large-scale cross-device FL with hundreds or thousands of clients. This should be added to the "Limitations" section.
>
> We extend our Conclusion and Limitation section to acknowledge that our experiments are limited to cross-silo settings and that further research is needed for cross-device scenarios. We motivate our choice of a small number of clients in the Methodology section when we first introduce the federated learning setting: we argue that a cross-silo setting is realistic for medical scenarios and discuss the computational constraints of cross-device experiments.
>
> > Refine the Theoretical Discussion: The hypotheses in Appendix L should be framed more clearly as speculative and be positioned as directions for future work.
>
> We fully agree with the reviewer that our theoretical discussion is speculative and mainly serves the purpose of suggesting directions for future work. We explicitly mention this limitation in the Conclusion section. As Reviewer 86LA suggested moving key points of the theoretical discussion to the main body, we created a second-to-last Section “Potential Theoretical Explanations” with our main hypotheses while keeping the discussion clearly framed as speculative.
>
> Thank you again for the constructive feedback and we hope that the revised manuscript addresses the reviewer's comments.
>
> [1] “Differentially private low-rank adaptation of large language model using federated learning”, Liu et al., ACM Transactions on Management Information Systems, 2024
>
> [2] “Analyzing leakage of personally identifiable information in language models”, Lukas et al., IEEE, 2023

---

### Review · Reviewer_MctV · 2025-11-10

**Summary Of Contributions:**

1.**Addresses a Timely Problem**- The paper tackles the central challenge of data privacy leakage resulting from unintended memorization in the impactful and growing domain of Federated Learning (FL) for Large Language Models (LLMs). This is crucial in the current era, where high-performing LLMs pose a direct risk of exposing sensitive client information, making the work immediately relevant to high-risk domains (medicine, law, finance).

2.**Impactful Empirical Finding**- The core finding—that Low-Rank Adaptation (LoRA) reduces unintended memorization by up to a factor of 10—is compelling because it is achieved with little to no reduction in model performance. This is a great finding because LoRA also significantly helps reduce communication cost (by sending only low-rank matrices) and fine-tuning cost (by only updating a small fraction of the parameters), offering a solution that is simultaneously a privacy enhancer and an efficiency accelerator in FL.

3.**Investigation of Multi-Layered Privacy Interactions**- The paper investigates the relationship between LoRA and other privacy-preserving techniques like Differential Privacy (DP) (gradient clipping/noising), Secure Aggregation, and Goldfish Loss.

4.**Extensive Experimental Design**- The study is scientifically sound, featuring a comprehensive set of experiments that ensures the robustness and generalizability of the findings across multiple scenarios.

**Additional Comments:**

The paper is well-written and easy to follow. The authors provide an extensive literature review that clearly positions the work within existing research.

**Audience:**

Yes

**Audience Explanation:**

Yes, I believe so. As discussed in the submission, the privacy implications of fine-tuning large language models (LLMs) are an active and rapidly developing research area. Recent works have begun to examine how different fine-tuning strategies affect privacy, with some focusing specifically on unintended memorization. Notably, a recent study has shown that LoRA can substantially reduce memorization risks compared to full fine-tuning in centralized learning settings. However, to the best of my knowledge, there has not yet been a systematic investigation of this question in the federated learning context. This makes the study particularly relevant, given that LoRA also offers practical advantages for FL — such as lower computational and communication costs — which further increase its potential impact and appeal to the TMLR community.

**Broader Impact Concerns:**

The paper discusses methods that reduce memorization and thereby improve privacy in LLM fine-tuning. While this is a positive contribution, as is often the case with privacy-oriented research, such techniques should be approached with some caution. For instance, reducing memorization can enhance data protection but may also make it more difficult to identify the use of copyrighted, restricted, or otherwise sensitive data in model training. It is therefore important to ensure that such methods do not create a false sense of security regarding data privacy or ethical compliance.

**Claims And Evidence:**

Yes

**Claims Explanation:**

1.Comprehensive Scope of Empirical Validation- The authors demonstrate the efficacy of LoRA through a broad experimental setup, which ensures the generalizability and practical relevance of their primary claim:

 - Real-World Relevance Across Domains: The study's focus on datasets from three high-risk applied domains—health, law, and finance—provides strong evidence that the privacy benefits of LoRA hold in sensitive, real-world deployment scenarios.

  - Scalability and Architectural Validation: They confirm the effect is consistent across a wide range of model architectures and scales (e.g., Llama and Mistral families, from 1B up to 70B parameters), validating that the findings are not model-specific.

   - Multi-Faceted Evaluation: The use of multiple, complementary metrics (Exact Match, BLEU, BERTScore) moves beyond simple token-level checks to rigorously capture both exact and approximate unintended memorization.

 - Sensitivity Analysis: The authors investigated the impact of key hyperparameter design choices, such as the LoRA rank (r) and the number of FL rounds, on model behavior.

- Security Risk Modeling: Testing under practical security challenges, including high data duplication (10x) and varying prompt lengths, confirms the method's resilience against common real-world attack vectors.

- Comparison against Centralized Learning: The paper offers a comparative analysis of memorization in Federated Learning (FL) versus Centralized Learning (CL) comparing both the effectiveness and the mechanics of LoRA's privacy enhancement.

2.Rigorous Investigation of Layered Privacy Defenses-The paper analyzes how LoRA integrates with and modifies the performance of other established privacy methods:

- Hyperparameter Sweeps: The combination of LoRA with each privacy method was studied by varying the main hyperparameters (HPs) involved (e.g., DP noise levels or Goldfish loss weight).

 - Impact of Stacking Defenses: The comparative results delineate effective combinations (e.g., LoRA with Goldfish Loss and DP Gradient Clipping) which further improve privacy, while highlighting the nuance of ineffective stacks (like LoRA with NEFTune).

**Requested Changes:**

Here are a few questions I would like to discuss with the authors:

Q1. The number of FL rounds (or local epochs) appears to be a critical hyperparameter, requiring a value high enough for utility but low enough to avoid unnecessarily increasing memorization. Given the cost, what strategy do the authors recommend for cost-effectively determining this optimal number of rounds?

Q2. Judging from Figure 3, it appears that for some larger models, Centralized Learning with LoRA (CL LoRA) achieves a lower unintended memorization rate than Federated Learning with LoRA (FL LoRA). Is it accurate that, in these specific scenarios, the immediate privacy concern (memorization) is better addressed by CL LoRA? If so, what is the authors' hypothesis for this finding, considering that FL is generally intended to be more privacy-preserving?

Q3. The paper's conclusion rests on LoRA providing a large privacy gain with minimal utility loss. While Table 4 (Law) and Table 5 (Finance) show evidence of memorization reduction, they do not report the primary task-specific utility metric (e.g., F1 score, Accuracy). Would you be able to add those?

Q4. The paper observed that the effectiveness of LoRA in reducing memorization varied notably between the Llama and Mistral model families. Given this variation, I was wondering why the experimental design heavily relies on multiple models within the Llama family (1B to 70B variants) instead of including models from fundamentally different open-source architectures available at the time?

---

> ### Author Response · Authors · 2025-11-23
> **Response to Reviewer MctV**
>
> Thank you for your positive assessment of our work.
>
> > Q1. The number of FL rounds (or local epochs) appears to be a critical hyperparameter, requiring a value high enough for utility but low enough to avoid unnecessarily increasing memorization. Given the cost, what strategy do the authors recommend for cost-effectively determining this optimal number of rounds?
>
> We agree that the number of FL rounds (or local epochs) is a critical hyperparameter. In practice, we recommend a cost-effective procedure based on privacy-aware early stopping:
>
> 1. Inject canary sequences, as described in Section 3.1, to obtain a scalable proxy for monitoring memorization during training.
> 2. Track memorization alongside utility, and trigger early stopping based on the privacy signal, rather than relying solely on validation accuracy.
>
> Our experiments show that most memorization occurs only after accuracy has already plateaued. This makes privacy-driven early stopping particularly effective: once the privacy signal begins to increase, further rounds tend to add memorization without improving utility, making this a natural and efficient point to stop training.
>
> > Q2. Judging from Figure 3, it appears that for some larger models, Centralized Learning with LoRA (CL LoRA) achieves a lower unintended memorization rate than Federated Learning with LoRA (FL LoRA). Is it accurate that, in these specific scenarios, the immediate privacy concern (memorization) is better addressed by CL LoRA? If so, what is the authors' hypothesis for this finding, considering that FL is generally intended to be more privacy-preserving?
>
> Indeed, in a few specific cases in the original Figure 3 (Figure 4 in the revised version), CL LoRA yields slightly lower unintended memorization than FL LoRA. However, this outcome is largely explained by the comparison setup rather than by an inherent privacy advantage of centralized training.
> In the original figure, the centralized model was evaluated at a single checkpoint after 3 epochs, whereas the FL curves showed memorization across many rounds, effectively corresponding to more training and thus more opportunities for memorizing. We also observed that, in a small number of instances, FL showed higher memorization even before reaching 3 rounds. Our hypothesis is that this arises from the non-IID client data, where local training on domain-specific distributions can temporarily increase memorization, while the aggregation step may not counteract this effect as effectively as centralized training.
> To avoid the misleading impression that the centralized model “stays stable” while the FL model evolves, we updated Figure 4 (formerly Figure 3) to compare only LoRA and full fine-tuning within the federated setting, and the CL–FL comparison is now more fairly reflected by examining Figures 2 and 3 together, where both settings are using a fixed number of epochs/rounds.
>
> > Q3. The paper's conclusion rests on LoRA providing a large privacy gain with minimal utility loss. While Table 4 (Law) and Table 5 (Finance) show evidence of memorization reduction, they do not report the primary task-specific utility metric (e.g., F1 score, Accuracy). Would you be able to add those?
>
> We designed our main medical scenario to account for potential utility loss and memorization on realistic sensitive data canaries. In contrast, our law and finance experiments have been designed to follow previous work (Carlini et al. 2023b, Ippolito et al 2023) which evaluates memorization indiscriminately on all the dataset without controlling for accuracy. As such, we did not include accuracy benchmarks either in our experiments design for these additional domains. Nevertheless, we controlled the validation loss of each fine-tuning to prevent overfitting.
>
> > Q4. The paper observed that the effectiveness of LoRA in reducing memorization varied notably between the Llama and Mistral model families. Given this variation, I was wondering why the experimental design heavily relies on multiple models within the Llama family (1B to 70B variants) instead of including models from fundamentally different open-source architectures available at the time?
>
> We focused on the Llama family because, at the time of experimentation, they were the only widely available and competitive open-source models across a broad range of sizes (1B–70B). Other architectures that later became strong baselines, such as Qwen, were not yet mature enough. We also tested Phi and SmoL models, but their utility in our FL and CL tasks was too low to yield meaningful privacy–utility comparisons. For these reasons, we concentrated on models that provided reliable results, while ensuring to include at least two different architectures to show that LoRA’s behavior is not uniform across architectures.

---

> > ### Comment · Reviewer_MctV · 2026-01-06
> >
> > I am happy with the paper being accepted

---

### Decision · Action_Editor_oqkG · 2026-01-24

**Recommendation:** Accept as is

**Additional Comments:**

The recommendation is based on the reviewers' comments, the action editor's evaluation, and the authors’ response.

This paper conducts a comprehensive analysis and evaluation of how LoRA can be used to reduce memorization during federated for LLMs.  =
All reviewers find the results valid and provide new insights. The authors’ rebuttal has successfully addressed the major concerns of reviewers. Therefore, I recommend acceptance of this submission.

**Audience:**

Yes

**Audience Explanation:**

Of broad interest to computer vision and parameter-efficient fine-tuning, machine unlearning, and fenerated learning

**Claims And Evidence:**

Yes

**Claims Explanation:**

The claims of mitigating unintended memorization in Federated Learning for LLMs using with LoRA are supported by the empirical results.